



# Two optimized methods for the quantification of anthropogenic and biogenic markers in aerosol samples using liquid chromatography mass spectrometry and gas chromatography mass spectrometry

Diana L. Pereira[1], Aline Gratien[1], Chiara Giorio[2], Emmanuelle Mebold[3], Thomas Bertin[4],
Cécile Gaimoz[4], Jean-François Doussin[4], and Paola Formenti[1]

[1]Université Paris Cité and Univ Paris Est Creteil, CNRS, LISA, Paris, F-75013, France,

[2]Yusuf Hamied Department of Chemistry, University of Cambridge, Cambridge, CB2 1EW, United Kingdom

[3]Univ Paris Est Créteil, CNRS, OSU-EFLUVE, Créteil, F-94010, France

[4]Univ Paris Est Creteil and Université Paris Cité, CNRS, LISA, Créteil, F-94010, France

*Corresponding authors:* diana.pereira@lisa.ipsl.fr, aline.gratien@lisa.ipsl.fr

**Abstract.** In this study, we present two optimized analytical methods for the quantification of molecular markers to attribute the contribution of various Volatile Organic Compounds (VOC) oxidation products to Secondary Organic Aerosol (SOA). Those involve Ultrahigh Performance Liquid Chromatography Electrospray Ionization coupled to Ion Mobility Time of Flight Mass Spectrometry (UPLC/ESI-IMS-QTOFMS) and Gas Chromatography Mass Spectrometry (GC-MS). Liquid extraction was performed for both techniques, with an extra derivatisation step with N,O-Bis(trimethylsilyl)trifluoroacetamide (BSTFA) containing 1 % trimethylchlorosilane (TMCS) for GC-MS analysis, enhancing the compound detection capacity. Between the two techniques, 23 biogenic and anthropogenic markers were identified, with five common species detected. Recoveries > 80% were observed for nitro-containing compounds and > 66% for aromatic and non-aromatic acids except for 3-methyl-1,2,3-butanetricarboxylic acid. Limits of detection < 5 ng were observed by UPLC/ESI-IMS-QTOFMS analysis for 4-nitrophenol and 2-methyl-4-nitrophenol, while GC-MS (with BSTFA derivatisation) analysis allowed better detection of lower mass compounds (for example limit of detection for 2-methylerythritol was 0.10 ng). While UPLC/ESI-IMS-QTOFMS allow the analysis of high molecular weight compounds at high resolution and sensitivity, GC-MS analysis focus on compounds of lower mass and higher polarity, together, these complementary methods provide a comprehensive tool for the quantification of organic markers arising from the airborne transformation of compounds of both biogenic and anthropogenic origins.

## 1 Introduction

Secondary organic aerosol (SOA) can contribute approximately 70% to the organic aerosol (OA) (Hallquist et al., 2009; Srivastava et al., 2018a) and influence both the Earth's climate and human health (Fan et al., 2022; Jimenez et al., 2009). Understanding its origin, and hence quantifying their sources it is essential for many source apportionment studies (Srivastava et al., 2018a). This implies detailed understanding on its molecular composition, particularly through the determination of organic molecular markers. While it is relatively well established for primary OA sources, it is more challenging for secondary OA sources, which result from



the oxidation of volatile organic compounds (VOC), as atmospheric oxidation processes lead to thousands of products varying

upon conditions, space and time. Proper markers quantification is henceforth needed to evaluate the contribution of specific VOCs to SOA chemical composition. Approaches such as the molecular markers method, positive matrix factorization and chemical mass balance are commonly used to achieve this (Srivastava et al., 2018a).

Molecular markers have to be both conservative and source specific (Nozière et al., 2015). Some have been identified for major biogenic and anthropogenic VOCs such as isoprene, pinene and aromatic oxidation products (Claeys and Maenhaut, 2021; Forstner

et al., 1997; Kleindienst et al., 2007; Sato et al., 2022). Isoprene photo-oxidation initiated by OH-radical has led to the formation and identification of polyol markers such as 2-methylthreitol, 2-methylerythritol (Claeys et al., 2004), and 2-methylglyceric acid (Edney et al., 2005). Pinene oxidation has been associated with a higher number of markers from first and second generation. For example, first oxidation products such as pinic acid, cis-pinonic, norpinic acid and terpenylic acid can derive from pinene ozonolysis reactions (Claeys et al., 2009; Yu et al., 1999). Further OH-oxidation of pinic and cis-pinonic acids involves the

formation of 3-methylbutane-1,2,3-tricarboxylic acid (MBTCA) (Szmigielski et al., 2007) and 3-Hydroxyglutaric acid (Claeys et al., 2007). Terpenylic acid oxidation by OH leads to terebic acid formation (Yasmeen et al., 2010). Evidence for the formation of organosulfates and nitrooxy organosulfates markers from biogenic VOCs has been also provided (Surratt et al., 2007, 2008).

A lower specificity is observed for common anthropogenic photo-oxidation products assigned as markers for aromatic compounds. 2,3-dihydroxy-4-oxopentanoic acid (DHOPA) has been considered the most common monoaromatic marker (Al-Naiema and

Stone, 2017; Kleindienst et al., 2007) while phthalic acid is linked to diaromatic compounds (Kleindienst et al., 2012). Aromatic compounds such as 4-hydroxy-3-methyl-benzaldehyde, 4-nitrophenol and 2-methyl-4-nitrophenol, p/m-toluic acid (Forstner et al., 1997), 2,5-dihydroxy benzoic acid (Hamilton et al., 2005), salicylic acid (Jang and Kamens, 2001) and open ring products such as glycolic acid (Kleindienst et al., 2004), succinic acid, and malonic acid (Sato et al., 2007) were also associated with aromatic VOCs oxidation.

Analytical tools that focus on the identification and quantification of organic markers in aerosol samples generally comprise the analysis of polar compounds with hydroxyl, carbonyl and carboxyl groups, normally achieved using chromatography-based methods (Nozière et al., 2015). Liquid Chromatography- Mass Spectrometry (LC-MS) and/or Gas Chromatography- Mass Spectrometry (GC-MS) are the most common techniques used in the identification of markers (Albinet et al., 2019; Iinuma et al., 2010; King et al., 2019; Kleindienst et al., 2004; Lanzafame et al., 2021; Sato et al., 2022; Srivastava et al., 2018b). They require

additional steps for sample extraction and, in the case of highly polar compounds, derivatization. Dichloromethane (Hu et al., 2008), acetonitrile aqueous mixtures (Yu et al., 1998), methanol and methanol/dichloromethane mixtures (Pashynska et al., 2002) are common extraction solvents for GC-MS while methanol and acetonitrile are the most common organic mobile phases used in LC-MS analysis (Gao et al., 2021; Grace et al., 2019; Hutchinson et al., 2012).

Protocols for the analysis of organic markers from aerosol particles have been previously developed (Albinet et al., 2019; Chien et

al., 1998; Hoffmann et al., 2007; Hu et al., 2008; Ikemori et al., 2019; King et al., 2019; Pashynska et al., 2002; Yu et al., 1998). Those protocols either focus on a specific type of marker or a group of them. For example, Pashynska et al. (2002) developed a GC-ion trap-MS method to follow levoglucosan and monosaccharides anhydrides as markers of biomass burning, with recoveries



>90%. Hoffmann et al. (2007) also followed biomass burning markers using high performance liquid chromatography/atmospheric pressure chemical ionisation mass spectrometry (HPLC/APCI-MS) with instrumental limits of detection (LODs) lower than 786.2 ng mL$^{-1}$ and recoveries > 15%. Ikemori et al. (2019) focus on the quantification of nitroaromatic compounds with LC-MS/MS analysis and polar acids by GC-MS with instrumental limit of detections (LODs) of 0.64 to 4.2 ng mL$^{-1}$ and 0.6 to 1 ng mL$^{-1}$, respectively. Recoveries were reported as > 90%. An LC-MS method for the detection of terrestrial and marine biomarkers (e.g., pinene, isoprene) in ice cores was developed by King et al. (2019). LOD varied between 2 to 20 ng mL$^{-1}$ with average recoveries of 80%. Yu et al.'s (1998) GC-MS method allows for the detection of biogenic and anthropogenic markers in the order of pg mL$^{-1}$ and 100 % was assumed as collection and derivatization efficiency. Albinet et al. (2019) provided a methodology comparing HPLC/MS-MS and GC-MS protocol development for target common biogenic and anthropogenic markers such as those associated with pinene, isoprene and aromatics oxidation. Compound-dependent limit of quantifications (LOQs) between 0.6 and 14.3 ng mL$^{-1}$ were reported for GC-MS and between 1.0 and 4.0 ng mL$^{-1}$ for HPLC/MS-MS. Recovery rates ranged between 10 and 90%.

In this work, we discuss two methods for the detection and quantification of varied biogenic and anthropogenic organic markers with their validation parameters. The methods were developed using Ultrahigh Performance Liquid Chromatography Electrospray Ionization coupled to Ion-Mobility Time of Flight Mass Spectrometry (UPLC/ESI-IMS-QTOFMS) and gas chromatography mass spectrometry (GC-MS). While UPLC/ESI-IMS-QTOFMS will allow the analysis of high molecular weight compounds at high resolution and sensitivity, the GC-MS analysis is focused on compounds of lower molecular weight. Together, these complementary methods will provide a comprehensive tool for the quantification of organic markers with different chemical functionalities including aromatic and nonaromatic compounds. The differences between both methods are evaluated in terms of their performance and compounds quantification in real atmospheric samples.

## 2 Materials and methods

The quantification of 23 organic molecular markers was performed by means of UPLC/ESI-IMS-QTOFMS and GC-MS. The steps undertaken to optimise the methods are described in the "Results and discussion" section.

### 2.1 Chemicals and reagents

All chemicals, gases and solvents used during the analysis of the organic markers are summarized in Table S1. The different compounds (Table 1) were selected as available commercial standards of common oxidation products of major VOCs precursors of biogenic and anthropogenic origin. The biogenic markers are α- and β-pinene oxidation products and isoprene oxidation products. The anthropogenic markers belong to the oxidation of different aromatic precursors such as benzene, toluene, ethylbenzene, xylene (BTEX) and naphthalene. Two of them are markers of biomass burning. The organic markers selected in this study belong to the list of commonly used markers of biogenic and anthropogenic origin implemented by the European Calibration Center (OGTAC-CC) (Herrmann and Mutzel, 2019; Mothes and Herrmann, 2024) of the ACTRIS research infrastructure (Laj et al., 2024).






**Table 1.** List of target analytes for this study representing molecular markers of biogenic and anthropogenic SOA.

| Class | precursor | Target marker |
|---|---|---|
| Biogenic | α-pinene | Cis-pinonic acid |
| | α/β-pinene | Pinic acid |
| | α/β-pinene | Norpinic acid |
| | α/β-pinene | Terebic acid |
| | α-pinene | 3-methyl-1,2,3-butanetricarboxylic acid (MBTCA) |
| | α-pinene | (1S,2S,3R,5S)-(+)-Pinanediol |
| | β-pinene | 1R-(+)-Nopinone |
| Biogenic | isoprene | α-methylglyceric acid |
| | | 2-methylerytritol |
| Biomass burning | Aromatics | 4-nitrocatechol |
| | | Syringaldehyde |
| Anthropogenic | Naphthalene | 4-methyl-phthalic acid |
| | | Phthalic acid |
| Anthropogenic | Aromatics | 2,3-dihydroxy-4-oxopentanoic acid (DHOPA) |
| | | 2,5-dihydroxy benzoic acid |
| | | Succinic acid |
| | | Glycolic acid |
| | | 3-acetyl-benzoic acid |
| | | salicylic acid |
| | | o-toluic acid |
| | | 4-nitrophenol |
| | | 2-methyl-4-nitrophenol |
| | | 2-hydroxy-3-methylbenzaldehyde |

## 2.2 Sample collection

In this work, PM$_1$ samples were collected on 150 mm diameter quartz fiber filters (Pallflex Tissuquartz), previously baked at 550

°C for 8 hours. After exposure, samples were conserved in pre-baked aluminum foil and sealed at -20 °C. Sampling was performed during daytime (6:00 – 22:00, local time) and night-time (22:00 – 6:00, local time) in the framework of the ACROSS (Atmospheric Chemistry Of the Suburban Forest) campaign (Cantrell and Michoud, 2022) at the forest of Rambouillet (France), in the summer 2022. An automatic continuous high-volume aerosol sampler (30 m$^3$ h$^{-1}$) DHA-80 (DIGITEL Enviro-Sense) was used. The procedures of aerosol sampling in the field are fully described in (Pereira et al., 2025). The samples discussed in this work were

collected on July 3, 4, 11, 12, 13, 17, 18 and 19.

## 2.3 UPLC/ESI-IMS-QTOFMS method

### 2.3.1 Sample extraction

Samples (punch of 46 mm) were spiked with 5 µL of the internal standard (1S)-(+)-camphor-10-sulfonic acid (Sigma Aldrich, 98%) at 20 µg mL$^{-1}$ in 50:50 (v/v) acetonitrile/ ultrapure water and cut into smaller pieces. Pieces were transferred to amber vials





and extracted with 4 mL of acetonitrile (ULC/MS-CC/SFC grade, Biosolve, 99.99%) using a Mini Shaker (15 mm Orbital, VWR) at 1000 rev min$^{-1}$ for 30 min. The extracts were individually filtered using a glass syringe coupled to a syringe filter (13 mm x 0.2 µm, VWR). The filtered solutions were then evaporated to dryness using a 12 positions N-Evap (Organomation) under a gentle stream of nitrogen at 99% purity fed by a CALYPSO 35 L min$^{-1}$ generator (F-DGSi, 2023). Samples were dissolved with 200 µL of 50:50 (v/v) acetonitrile/ ultrapure water, transferred to 250 µL vial inserts and stored for up to 24 h at -18°C prior to analysis.

**2.3.2 Analysis**

Samples were analysed by means of an UPLC/ESI-IMS-QTOFMS system consisting of an ACQUITY$^{TM}$ UPLC I-Class system and a Vion$^{TM}$ ion mobility hybrid (IMS) QTOF mass analyser (Waters$^{TM}$). A UPLC BEH C18 column (1.7 µm, 2.1x100 mm, Waters) was used as stationary phase.  Mobile phases were ultrapure water with 0.1 % formic acid (v/v) (A) and acetonitrile with 0.1 % formic acid (v/v) (B) with an elution gradient of: 2 min 5 % B, 2-32 min from 5 to 60 % B, 32-35 min from 60 to 95 % B,

35-38 min hold at 95% B, 38-40 min from 95 to 5 % B, and finally stabilization at 5 % B for 5 min. Separation was performed at a 40 °C column temperature and a flow rate of 0.4 mL min$^{-1}$. 2 µL of sample were injected in triplicates. Solvent blanks (50:50 (v/v) acetonitrile/ ultrapure water) were injected between replicates to check for any carry-over.

Analysis was performed in negative ionization mode with the following ESI parameters: 120 °C source temperature, 600 °C of desolvation temperature and cone gas flow of 150 L/h. Mass spectra were recorded in full scan mode in the $m/z$ range of 50-1000,

where the $m/z$ corresponds to the mass of the deprotonated molecules. Further data processing was performed only for a $m/z$ range between 50-350. Compounds were then identified by means of their $m/z$, retention time (Rt) and collision cross section (CCS) with $m/z$ error ≤ 5 ppm, CCS error ≤ 2% and Rt error ≤ 0,1 min. Quantification was done on the full scan spectra, in extracted ion current, with an external calibration in the range 10 to 200 µg mL$^{-1}$ with (1S)-(+)-camphor-10-sulfonic acid as internal standard. Calibrations were performed using the same extraction procedure as the real samples. Further details of the calibration procedure

are discussed in Section 3.1.5.

**2.4 GC-MS method**

**2.4.1 Sample extraction and derivatisation**

Samples (punch of 46 mm) were spiked with 5 µL of heptanoic acid (Sigma Aldrich, 99%) solution at 40 µg mL$^{-1}$ as internal standard and extracted in acetonitrile (HPLC grade, VWR, 99.95%), filtered and evaporated to dryness similarly to the extraction

procedure described in Section 0. Samples were then reconstituted by adding 50 µL of acetonitrile. 200 µL of N,O-Bis(trimethylsilyl)trifluoroacetamide (BSTFA) containing 1 % trimethylchlorosilane (TMCS) (Sigma Aldrich, purity 99 %) was added to each solution and then heated at 60°C for 30 min to allow derivatization. This derivatization process, including the volumes, was selected following Albinet et al. (2019). Final extracts were stored at -18°C and analysed as quickly as possible after 24 h.

Furthermore, derivatization was used as a support for product identification. In the derivatisation process with BSTFA, the labile hydrogens of the alcohols and acid functions of the compounds are replaced by thrimethylsilyl -Si(CH$_3$)$_3$ groups as follows:



Alcohols/acids     BSTFA

BSTFA derivatization gives specific ion fragments in mass spectra at m/z 73 $[Si(CH_3)_3]^+$ and 117 $[COO=Si(CH_3)_3]^+$. For compounds bearing labile H atoms, they exhibit a *m/z* 147 corresponding to $[(CH_3)_2Si=OSi(CH_3)_3]^+$. For identification of individual compounds using authentic standards and evaluation of their response, a 50:50 (v/v) mixture of BSTFA and the standard solution was left reacting overnight at room temperature.

### 2.4.2 Analysis

Analysis was performed using a GC-MS made of a gas chromatograph (Clarus 650, Perkin Elmer) and a mass spectrometer (MS SQ8C, Perkin Elmer). Separation was achieved using an analytical column RXi-5Sil MS (30 m, 0.25 mm ID, 0.25 µm Restek) and the following GC oven temperature: an initial temperature of 60°C hold during 15 min, followed by a ramp of 5°C min$^{-1}$ to from 60°C to 280°C, and 7 min at 280 °C. Helium was used as carrier gas at a flow of 1 mL min$^{-1}$. An injection volume of 5 µL was added using a high precision syringe (84301 CR700-20 1-20ul, Hamilton). Ionisation was performed using an electron impact source. Mass spectrometry analysis was performed in a mass range between 50 and 500 *m/z* with a mass scan time of 0.3 sec between 6 to 66 min. We used the full scan total ion current (TIC) mode and when additional compound verification was required, the selected ion recording (SIR) mode for 6 channels. Compounds were identified by monitoring the Rt and major or specific ions derived from fragmentation using a mass spectra database built from the individual standards injection. Some of them were also verified using the NIST (National Institute of Standards and Technology) library. Quantification was performed by ion mass extraction in the TIC mode for characteristic fragments, subtracting the signal of the blanks and using external calibration in the range 10 to 200 µg mL$^{-1}$ with heptanoic acid as internal standard. Calibration standards underwent the same extraction and derivatisation procedure as the real samples. Further details are provided in Section 3.2.3.

### 2.5 Cleaning procedure between experiments

In between experiments, all the glassware used was cleaned in an automatic laboratory glasswasher PG 8593 [WW AD] (Miele) following a protocol for organic resides cleaning (Miele, 2022). Cleaning was performed with tap water at 75 °C using a KOH solution neodisher® LaboClean FLA as detergent. Then, the glassware was rinsed with a $H_3PO_4/C_6H_8O_7$ solution neodisher® N as neutralizer in distilled water, followed by a second rinsing cycle in distilled water at 75°C and dried at 110 °C during 30 min. Afterwards, the material was covered with aluminium foil and baked in a furnace at 500 °C for 2 hours to remove possible additional organic contaminants.

The metallic tips used for evaporation were cleaned with an ultrasonic bath for 15 min using isopropanol. Test and final calibrations were performed on quartz filters, which were cleaned by baking at 550 °C for 8 hours and stored in pre-beaked aluminium foil under a laminar flow hood. For the UPLC/ESI-IMS-QTOFMS, the ESI source cone was manually cleaned between experiments



using aluminium oxide powder (Restek) and sonicated for 10 min in an ultrasonic bath first with ultrapure water and then with isopropanol (LC-MS grade).

**2.6 Method validation**

The performances of both methods were assessed by analysing the following analytical parameters: variability (Eq. 1), sample recovery (Eq. 2), the coefficient of variation of the method (Eq. 3) and LOD. The variability of measurements for UPLC/ESI-IMS-QTOFMS analysis was obtained by calculating the percent variability between triplicate injections as:

$$\textbf{Variability } (\%) = \frac{\textbf{standard deviation}}{\textbf{sample mean}} \cdot \textbf{100} \tag{1}$$

The recovery for each compound for both methods was calculated using the ratio between the amount of the compound found after extraction and the amount added to a filter blank before extraction as:

$$\textbf{Recovery } (\%) = \frac{\textbf{mass recovered}}{\textbf{mass deposited}} \cdot \textbf{100} \tag{2}$$

Following the ISO 8466-1:2021 standard, the coefficient of variation of the method ($V_{XO}$) for the target compounds was calculated using Equation 3:

$$\textbf{V}_{\textbf{xo}} \ (\%) = \frac{\textbf{S}_{\textbf{y}}}{\textbf{b} \cdot \bar{\textbf{x}}} \cdot \textbf{100} \tag{3}$$

The residual standard deviation ($S_y$) was calculated in function of the fit, using Eq. 4 for linear calibrations ($y=bx$) and Eq. 5 for quadratic functions ($y=a+bx+cx^2$) with a, b and c as calibration coefficients, $\bar{x}$ as the mean value of the different $x_i$ and $n$ the number of points considered in the calibration:

$$\textbf{S}_{\textbf{y}} = \sqrt{\frac{\sum_{i=1}^{n}[y_i - (b.x_i)]^2}{n-2}} \tag{4}$$

$$\textbf{S}_{\textbf{y}} = \sqrt{\frac{\sum_{i=1}^{n}[y_i - (a+b.x_i+c.x_i^2)]^2}{n-3}} \tag{5}$$

The LOD for each technique was compound specific for real standard solutions. For UPLC/ESI-IMS-QTOFMS, LOD for each compound was assigned as the x-value associated to the y-intercept of the confidence interval derived from the calibration curves using a linear fit with 95% confidence (Hubaux and Vos, 1970). For GC-MS, LODs were calculated as the blank response (from the calibrations) plus three times its standard deviation. All LODs reported here are in mass (in ng) of analyte referred to the whole filter sample (corrected for the size of the portion analysed). For comparison with the literature, LODs are additionally reported as the concentration in the injected solution (in ng mL$^{-1}$).

The two analytical methods were compared by quantifying compounds that are targets in both techniques on samples collected at the Rambouillet forest during the ACROSS campaign. Bland-Altman plots (Bland and Altman, 1995, 1999) were used for the comparison. Those consider the similarity of two independent methods by visual inspection of the difference in concentrations derived from GC-MS and UPLC/ESI-IMS-QTOFMS in function of the mean of both measurements.





## 3 Results and discussion

### 3.1 Optimisation of the UPLC/ESI-IMS-QTOFMS analytical method for the quantification of biogenic and anthropogenic markers

#### 3.1.1 Optimisation of the chromatographic method

**Selection of the chromatographic method**

Different methods were tested to separate and identify 14 compounds using the UPLC/ESI-IMS-QTOFMS: cis-pinonic acid, pinic acid, norpinic acid, terebic acid, MBTCA, 4-nitrocatechol, syringaldehyde, 4-methyl phthalic acid, phthalic acid, 3-acetylbenzoic acid, 4-nitrophenol, 2-methyl-4-nitrophenol, azelaic acid and (1S)-(+)-camphor-10-sulfonic acid. Initial methods tested on individual standard solutions at 10 µg mL⁻¹ in 100% methanol used elution gradients with varying mobile phases, slopes and lengths of 17 min and 60 min. Mobile phases consisted of 0.1 % formic acid (v/v) in ultrapure water (A) and methanol (B). For the 17 min elution method, the following gradient was used: 2 min isocratic at 5% B, 2-10 min linear gradient from 5 to 50% B, 10-11 min linear gradient from 50 to 99% B followed by an isocratic at 99% B for 2 min, 13-15 min linear gradient from 99 to 5% B and equilibration at 5% B for two min. For the 60 min method, the elution conditions were: 3 min isocratic at 5% B, 3-25 min linear gradient from 5 to 50% B, 25-43 min linear gradient from 50 to 90% B, 43-48 min linear gradient from 90 to 5% B and equilibration at 5% B for 3 min. The 17 min method showed the overlap and irregular peak shapes for MBTCA, terebic acid, and phthalic acid with coelution of pinic acid and 4-nitrophenol, and of norpinic acid and camphor sulfonic acid (Fig. S1). The 60 min method provided better performance associated with a decrease in the number of compounds coeluted (Fig. S2), however, given that most of the target compounds elute in the first 20 min, such a long method would be unnecessarily time and resource consuming.

An intermediate method of 45 min with mobile phases with 0.1 % formic acid (v/v) in ultrapure water (A) and acetonitrile or methanol as organic solvent (B) and the following gradient was tested: 2 min isocratic at 5% B, 2-32 min linear gradient from 5 to 60% B, 32-35 min linear gradient from 60 to 95% B, 35-38 min isocratic at 95% B, 38-40 min linear gradient from 95 to 5% B, and equilibration for 5 min at 5% B. Example chromatograms using the 45 min method with methanol and with acetonitrile as mobile phase are shown in Fig. S3. As observed, most of the standards were properly identified after chromatographic separation with good peak shapes and compounds eluting in the first 18 min for both organic solvents. Therefore, the 45 min elution method was selected for the analysis.

**Selection of the organic solvent for the chromatographic method**

The use of methanol and acetonitrile as organic solvents for the chromatographic separation was evaluated by comparing the compound responses obtained using both solvents as shown in Figure 1. As the use of 100% organic solvent to prepare the standard solutions under analysis may negatively influence the peak shape and thus prevents proper quantification, we selected 50/50 ultrapure water/organic solvents for the preparation of standard solutions. Among the 14 target compounds, six (pinic acid, 3-acetyl benzoic acid, 4-methyl phthalic acid, cis pinonic acid, syringaldehyde and 2-methyl-4-nitrophenol) showed a similar response with both solvents. In the case of terebic acid, camphor sulfonic acid, MBTCA, norpinic acid, and phthalic acid, higher responses were

observed with acetonitrile as organic solvent and the opposite effect was observed for azelaic acid and 4-nitrocatechol. Higher compound response using acetonitrile was considered when the variability between the responses for both solvents was >30%. Azelaic acid was initially consider as a target standard to identify possible sample contamination as it was identified by the software

at multiple Rt. After its identification, we discarded azelaic acid presence. Because of the higher elution power, lower pressure at a column temperature of 40 °C, and higher solubility for most of the compounds under analysis, resulting in a better compound response, the use of acetonitrile appears as the most suitable for the selected target compounds.

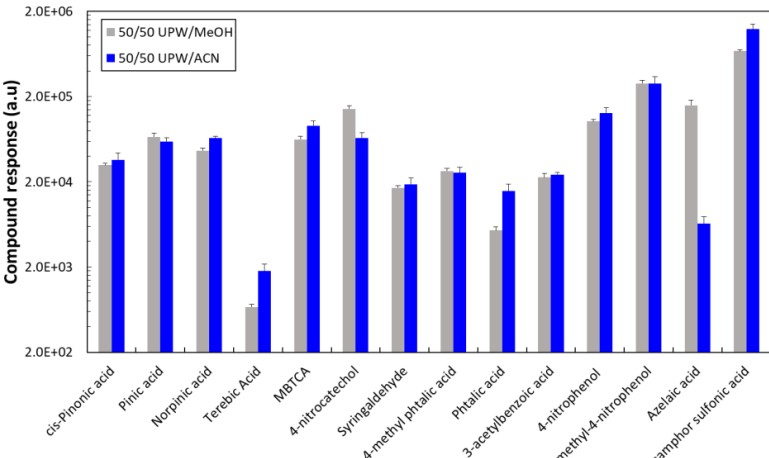

**Figure 1.** Comparison of the compound responses observed for the 45 min elution method. Individual standard solutions were prepared in 50/50 ultrapure water (UPW)/and organic solvent methanol (MeOH) or acetonitrile (ACN) at a concentration of 10 µg mL$^{-1}$.

### 3.1.2 Evaluation of the instrumental response with and without filter extraction

Samples in this work were collected on quartz fiber filters (Section 2.2), therefore, to mimic the sample conditions, we evaluate

the response of the compounds after performing the sample extraction procedure (Section 2.3.1). Two mixtures of compounds were prepared: biogenic (cis-pinonic acid, pinic acid, norpinic acid, terebic acid and MBTCA) and anthropogenic (4-nitrocatechol, syringaldehyde, 4-methyl phthalic acid, phthalic acid, 3-acetylbenzoic acid, 4-nitrophenol, 2-methyl-4-nitrophenol) mixtures at 400 µg mL$^{-1}$. Then, blank filters were spiked with 5 µL of a (1S)-(+)-camphor-10-sulfonic acid and 5 µL of each 400 µg mL$^{-1}$ mixture containing the compounds under analysis. As shown in Figure 2, the instrumental variability, between three replicates

randomly injected, of the mixture solution without filter extraction is less than 21% for all the target compounds. After filter extraction, a higher replicate variability for phthalic acid (27%), 4-methyl phthalic acid (27%), MBTCA (53%) and camphor sulfonic acid (35%) was observed. The response variability of the target compounds can result from the extraction procedure. Given the different compound polarities and volatilities, their extraction will be influenced by the dissolution, filtration and solvent evaporation steps.






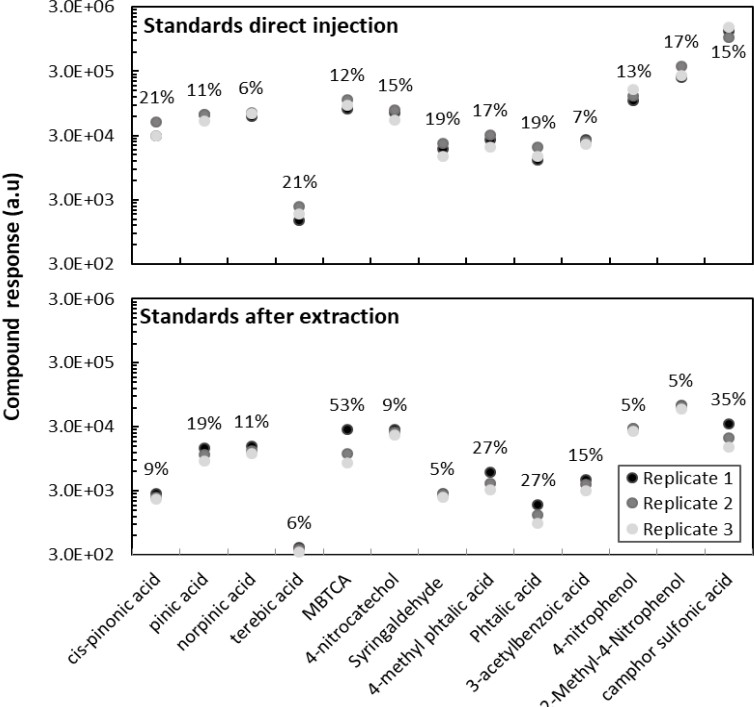

**Figure 2.** Standards injection response with and without filter extraction for various anthropogenic and biogenic compounds from a mixture at 400 µg mL$^{-1}$. Coefficients of variation are reported in the graph in percentages.

### 3.1.3 Optimization of the experimental setup (inserts and needle position)

Individual standards and liquid samples injected without extraction were conserved in inactinic glass vials of 1.5 mL with a solution volume of 0.5 mL. In the case of samples extracted from filters, a lower volume was used (200 µL) to increase compound concentrations and detection probability, requiring the use of vial inserts. In this study, we chose a conical bottom shaped insert with plastic spring. To understand possible artefacts from the experimental setup, we evaluated the needle position, effect of inserts and storage time influence on the sample stability on a mixture containing the target anthropogenic standards (Table 2). The needle

position test refers to variations in the height from the bottom of the vials.

We first focused on the analysis of anthropogenic standards in solutions containing mixtures at 0.5, 1.0 and 2.5 µg mL$^{-1}$. As shown in Fig. S4, the variability between replicates for samples directly injected from the vials is high (> 14 %). The compound-dependent injection variability from the vials was lower than 20% for syringaldehyde while for 2,5-dihydroxybenzoic acid it was much higher (87%). Signals were also higher than the one using inserts, for which the replicate variability is lower. These differences between

injections without and with inserts can be influenced by sample degradation and signal loss over time as samples in vials (without



inserts) were analysed first in the sequence. Although the differences between the use and absence of inserts, their presence is needed to maximize the sample intake into the system due to the low sample volumes after extraction. Additional variability at using inserts can derive from the formation of bubbles at the bottom of them. The presence of bubbles can reduce the amount of sample analysed and henceforth the response. This was discarded as changing the distance of the needle from the bottom of the vial between 5 and 10 mm did not affect the signals. As no influence was observed at varying the needle position, we selected 10 mm distance from the bottom for future experiments as a default parameter suggested by the manufacturer.

### 3.1.4 Optimisation of the system and instrumental response evaluation

A control solution containing a mixture of acetaminophen, leucine enkephalin, sulfadimethoxide, sulfaguanidine and Val-Tyr-Val (SST solution in the following text, Waters) was systematically used as quality control, injected in 5 replicates at the beginning of each experiment, to assess the quality of the instrument calibration. The same SST sequence was injected before and after a 34 injections sequence of samples (duration approx. 25 h) to monitor the evolution of signal intensity over time. A SST signal decrease by more than 70% was observed after the analysed sequence (Fig. S5), which was associated with an instrumental signal loss since the SST solution is normally stable for up to several months, and is thus unlikely degraded by 70% over a 25h period.

To follow the signal decrease more precisely over time, we additionally used the LockMass solution (Waters) which is infused continuously at 15 µL min$^{-1}$ to correct for minor mass deviations in real time throughout whole analysis sequences. The LockMass solution consists of 200 pg µL$^{-1}$ Leucine-Enkephalin in 50/50 acetonitrile/ultrapure water with 0.1% formic acid; this solution is normally stable for weeks at ambient temperature. For simple injection-to-injection comparison, the response of the lock mass was extracted by summing the signal (TIC) over the 45 min of each individual infusion (Fig. S6) and we observed a linear decrease in the solution response. Such variation observed at testing the signal evolution overtime for the SST and LockMass solutions could only derive from the instrumental signal loss overtime. Therefore, calibrations overtime where performed during analysis to account for the signal stability.

Additionally, the cone gas flow was tested at three levels to increase the signal stability: 50, 100 and 150 L h$^{-1}$ using a fresh mixture at 2.5 µg mL$^{-1}$ containing the target anthropogenic compounds. As shown in Table S2, similar responses were observed at different levels of cone gas flow for acids. However, at 150 L h$^{-1}$ a slightly higher response for nitrophenol compounds was observed. Therefore, 150 L h$^{-1}$ was selected as the cone gas flow for further analysis.

### 3.1.5 Calibration curves for target compounds

To evaluate the compound response and account for instrumental signal loss, calibrations were performed in sequences on non-consecutive triplicates, using increasing concentrations of mixtures containing the target compounds. Standard solutions containing the biogenic and anthropogenic mixtures of target compounds were prepared in 50/50 ultrapure water/acetonitrile at concentrations of 4, 10, 20, 40, 100, 150 and 200 µg mL$^{-1}$. For each standard solution, 5 µL of each standard mixture were individually added to clean (baked) quartz filters together with 5 µL of a solution of (1S)-(+)-camphor-10-sulfonic acid as internal standard. The filters were extracted following the procedure of Section 2.3.1. Calibration curves were performed by considering the mass of the compound deposited on the filter before extraction.



The variability of the response of the internal standard is represented in Fig. S7. A 60% decrease in the signal of camphor-10-sulfonic acid was observed between the first and second replicate, with a higher variability (30%) compared to the replicates 2 and 3 which had a response difference of 5%. A signal decrease from the first replicate was also observed for most of the calibration standards (Fig. S8). As an instrumental signal decrease was observed, the variations in the compound response between replicates can be attributed to this loss. For the nitrophenol compounds (2-methyl-4-nitrophenol, 4- nitrocatechol and 4-nitrophenol), an apparent increase in the signal response for intermediate concentration was observed. This variability can be influenced by the stability of the solution, possible signal interferences for nitro compounds and matrix effects which can affect individual compound responses, as this behaviour was not observed for acids. Carry-over effect on the column was rejected as an explanation as these nitro compounds were not detected in the blanks that were analysed between replicates.

At normalizing the compounds response for the internal standard response, the variability between the replicates decreased for compounds such as 4-methyl phthalic acid, but not for the nitro compounds. Although (1S)-(+)-camphor -10-sulfonic acid (open ring-acid) may not represent the more suitable internal standard for aromatic compounds, this acid shows variations < 30% being representative of most of the target compounds. Therefore, we use the normalization to its response to reduce the uncertainties associated with the system set-up and extraction procedure for the samples analysed here.

Linear calibrations with high determination coefficients were derived for all target compounds, including those with lower responses. We used sequences of a maximum of 54 injections (approx. 40 h). Because the quantification is replicate-dependent, for each set of experiments, we performed calibrations and sample injection in a consecutive way for each replicate. Calibrations overtime in the sequence account for instrumental signal degradation and allows to perform the quantification with the closest replicate. The calibration for each replicate is reported Table 2, together with the *m/z*, Rt and CCS values used to identify each compound. For compounds for which the determination coefficient of the linear regression ($R^2$) was lower than 90%, quantification was performed just with two of the replicates to decrease the variation. In such cases, the third replicate was not considered due to signal loss overtime. $V_{XO}$ values for individual calibration replicates showed the increase in the method variability between replicate for compounds such as 2,5-dihydroxy benzoic acid (from 25% to 40%), which also showed the lowest recovery. The rest of the compounds, with exception of MBTCA, were associated to $V_{XO}$ values < 30%, showing the good performance of the method at considering linear calibrations. The final individual compound concentrations are reported as the arithmetic mean and the error associated to their concentrations consider the standard deviation between triplicates and the volume deviation during the sampling.





**Table 2.** Mass to charge ratio ($m/z$), collision cross section (CCS), retention time (Rt), limit of detection (LOD), recovery and calibration information (slope, $R^2$ and $V_{XO}$) for individual replicates (R1, R2 and R3). Calibrations represent the normalized response (compound response to the internal standard response) versus the mass deposit on the filter. Calibration curves were performed using camphor-10-sulfonic acid as internal standard at 20 µg mL⁻¹ for mixtures of anthropogenic and biogenic standards at concentrations between 10 to 200 µg mL⁻¹. $V_{XO}$ shows the coefficient of variation of the method for each replicate and b the slope.

| Organic compound | $m/z$ | CCS (Å²) | Rt (min) | LOD (ng) | Recovery (%) | CALIBRATION (normalized response vs mass on the filter) for each replicate | | | | | | | | |
|---|---|---|---|---|---|---|---|---|---|---|---|---|---|---|
| | | | | | | $b_{R1}$ (x10⁻⁴) | $R^2$ | $V_{XO,1}$ (%) | $b_{R2}$(x10⁻⁴) | $R^2$ | $V_{XO,2}$ (%) | $b_{R3}$(x10⁻⁴) | $R^2$ | $V_{XO,3}$ (%) |
| Cis-pinonic acid | 183.102 | 144.6 | 9.13 | 63 | 109 ± 15 | 0.54 ± 0.02 | 0.98 | 19 | 0.62 ± 0.02 | 0.99 | 12 | 0.61 ± 0,04 | 0.96 | 20 |
| Pinic acid | 185.082 | 137.9 | 6.23 | 44 | 123 ± 31 | 4.5 ± 0.3 | 0.93 | 28 | 5.1 ± 0.3 | 0.94 | 26 | 5.0 ± 0.3 | 0.96 | 21 |
| Norpinic acid | 171.066 | 134.1 | 4.90 | 38 | 109 ± 20 | 5.1 ± 0.2 | 0.97 | 17 | 5.9 ± 0.3 | 0.97 | 17 | 5.6 ± 0.2 | 0.98 | 15 |
| Terebic acid | 157.051 | 130.6 | 2.85 | 49 | 111 ± 26 | 1.6 ± 0.1 | 0.96 | 16 | 2.1 ± 0.1 | 0.96 | 18 | 2.2 ± 0.1 | 0.97 | 22 |
| MBTCA | 203.056 | 135.4 | 3.28 | 51 | 84 ± 10 | 4.4 ± 0.3 | 0.93 | 36 | 3.4 ± 0.3 | 0.91 | 29 | 3.4 ± 0.3 | 0.93 | 33 |
| 4-nitrocatechol | 154.015 | 120.3 | 6.19 | 32 | 80 ± 25 | 16.4 ± 0.9 | 0.96 | 24 | 24 ± 1 | 0.97 | 18 | 27 ± 2 | 0.97 | 17 |
| syringaldehyde | 181.051 | 135.1 | 7.10 | 140 | 92 ± 21 | 0.51 ± 0.04 | 0.87 | 23 | 0.74 ± 0.05 | 0.90 | 21 | 0.84 ± 0.07 | 0.86 | 16 |
| 4-methyl phthalic acid | 179.035 | 131.6 | 7.80 | 30 | 106 ± 18 | 1.9 ± 0.1 | 0.98 | 15 | 2.0 ± 0.1 | 0.99 | 11 | 2.0 ± 0.1 | 0.98 | 11 |
| Phthalic acid | 165.019 | 124.5 | 4.58 | 44 | 99 ± 11 | 0.59 ± 0.02 | 0.97 | 16 | 0.57 ± 0.02 | 0.98 | 11 | 0.53 ± 0.03 | 0.93 | 23 |
| 2,5-dihydroxy benzoic acid | 153.019 | 122.6 | 3.10 | 52 | 66 ± 10 | 1.5 ± 0.1 | 0.96 | 25 | 1.6 ± 0.1 | 0.93 | 40 | 1.5 ± 0.1 | 0.96 | 80 |
| 3-acetyl-benzoic acid | 163.040 | 133.7 | 8.10 | 36 | 117 ± 21 | 2.0 ± 0.1 | 0.95 | 21 | 2.4 ± 0.1 | 0.97 | 16 | 2.4 ± 0.1 | 0.98 | 15 |
| salicylic acid | 137.024 | 118.4 | 8.81 | 23 | 107 ± 19 | 1.7 ± 0.1 | 0.98 | 15 | 2.0 ± 0.1 | 0.97 | 17 | 2.0 ± 0.1 | 0.96 | 16 |
| 4-nitrophenol | 138.020 | 119.1 | 8.61 | 4.4 | 123 ± 32 | 15.1 ± 0.6 | 0.97 | 17 | 23 ± 1 | 0.98 | 20 | 26 ± 2 | 0.95 | 23 |
| 2-methyl-4-nitrophenol | 152.035 | 124.6 | 13.16 | 3.3 | 113 ± 18 | 33 ± 1 | 0.98 | 15 | 51 ± 2 | 0.98 | 13 | 60 ± 3 | 0.97 | 17 |



## 3.2 GC-MS method development for the quantification of molecular markers

GC-MS analysis with an extra derivatisation step with BSTFA was performed to evaluate the response of highly polar compounds and semi-volatile compounds. The GC-MS system used in this work is normally operated with supercritical fluid extraction (SFE) by using $CO_2$, as described in detail in Chiappini et al. (2006). The online SFE-GC-MS has been previously used for the quantification of biogenic (Chiappini et al., 2006) and anthropogenic hydrocarbons (Lamkaddam et al., 2020) and aromatic alkenes (Chiappini et al., 2019). In this work the online SFE extraction procedure described in the original protocol was substituted by liquid extraction of the samples and direct injection in the system.

### 3.2.1 Evaluation of the target compounds' response

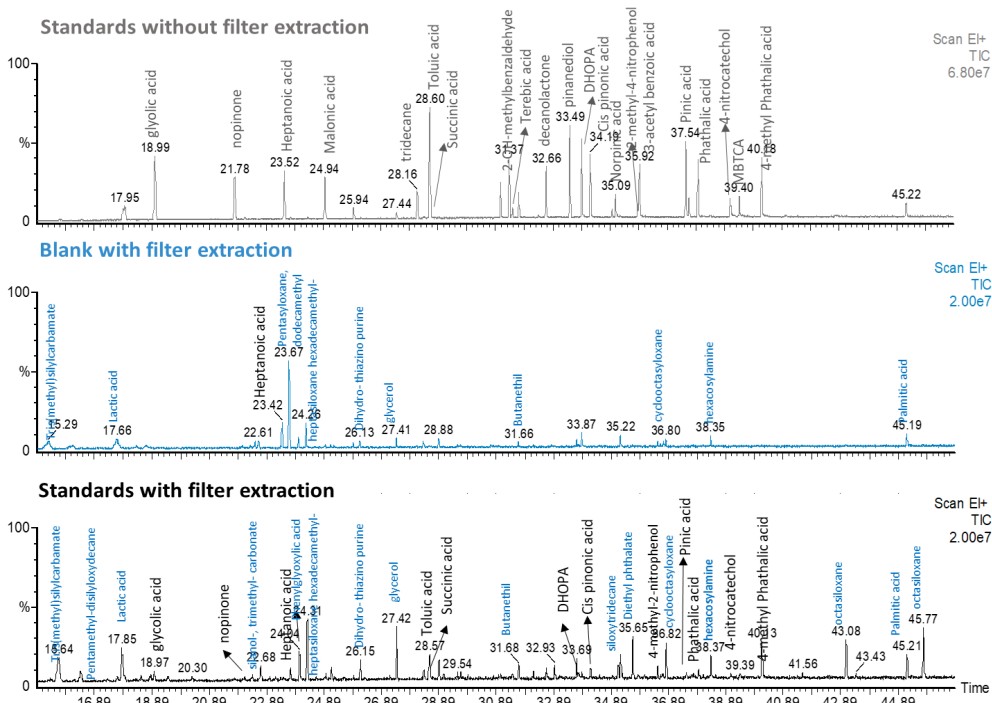

**Figure 3.** TIC Chromatogram of a mixture of anthropogenic and biogenic compounds injected directly in the GC-MS after derivatisation without filter extraction (top panel) and with filter extraction (bottom panel). The middle panel accounts for the blank filter extraction. In the bottom panel, peak assignation for the standards compounds are highlighted in black colour, while blue represent the peak identification performed with the NIST library.



Compound response was evaluated for a mixture of 19 biogenic and anthropogenic markers (Table S1) with and
without filter extraction. First, after an over-night derivatisation as described in section 0, a mixture of the target
compounds at 6 µg mL$^{-1}$ was directly injected in the system on the following day. All the compounds added to the
mixture were detected (Figure 3). Therefore, 10 µL of a standard mixture of the target compounds at 50 µg mL$^{-1}$ (more
concentrated to account for dilution) was spiked on a blank filter together with the internal standard and extracted as
described in Section 0. A blank filter containing only the internal standard was simultaneously extracted to identify
possible interferences.

As shown in Figure 3, there was an increase in the background signal represented by a blank filter (blue chromatogram)
compared to the standard solution directly injected without any extraction step, but with a derivatisation step. Some of
the peaks observed in the blank were also present in the filter containing the standards mixture. Those were assigned
following the NIST library and most of them were identified as Si-groups such as siloxane and silanol compounds,
which can come from impurities present in the quartz filters that appear during the extraction. In contrast to direct
injection, not all the compounds (malonic acid, terebic acid) present in the solution were identified, showing possible
issues during the extraction procedure and/or derivatisation time. It is worth to consider that after extraction, the
response of compounds of higher volatility can be influenced by the solvent evaporation step. Another additional peaks
can arise from experimental manipulation (e.g., palmitic acid, lactic acid) or impurities present in the quartz filters
used.

### 3.2.2 Analysis of blank contributions from the experimental procedure

As a significant blank contribution was observed, we tested a mixture of acetonitrile HPLC grade (VWR chemical,
99.95% purity) and the derivatisation reagent BSTFA with the different steps of the method. First, the mixture was
directly injected into the system and the noise level of Figure 3 or possible impurities were not observed (grey plot,
Fig. S9). Consecutively, the solution containing both components (solvent and BSTFA) was heated following the
derivatisation protocol, and the peaks previously observed in the blank in Figure 3 were also present (blue plot, Fig.
S9). As this blank contribution could originate from possible impurities derived from the solvent-filter interaction, we
evaluated the response of acetonitrile ULC/MS-CC/SFC grade (Biosolve, 99.99% purity) as it has a higher purity. No
improvements in the blank signals were observed by switching the solvent (black plot, Fig. S9).

The derivatisation procedure is required to allow the decrease in polarity of some target compounds and therefore their
identification and quantification. As signal contributions from the solvent, quartz fiber filters and derivatization process
cannot be avoided and most of the target compounds can be identified, calibrations were performed with those
conditions. Therefore, during the extraction of samples, a blank filter was simultaneously analysed, and its contribution
was subtracted from the samples.





### 3.2.3 Evaluation of calibration

As deuterated standards were not commercially available, we selected heptanoic acid as internal standard. In online SFE-GCMS, tridecane and/or o-toluic acid have been used as internal standards (Chiappini et al., 2006, 2019; Lamkaddam et al., 2020). Here, heptanoic acid was selected over other compounds such as tridecane and octanal due to its higher solubility in acetonitrile and presence of a labile proton, making it the more representative of the target compounds (mainly carboxylic acids). The variability between replicates for the heptanoic acid was evaluated with and without filter extraction as shown in Fig. S10. A minimum variability was observed for the peak area without filter extraction (< 3%), while it reached 36% for solutions extracted in different days. A similar variability between samples and replicates was also observed for octanal (30%), and attributed to the extraction procedure as no influence of the derivatisation was found.

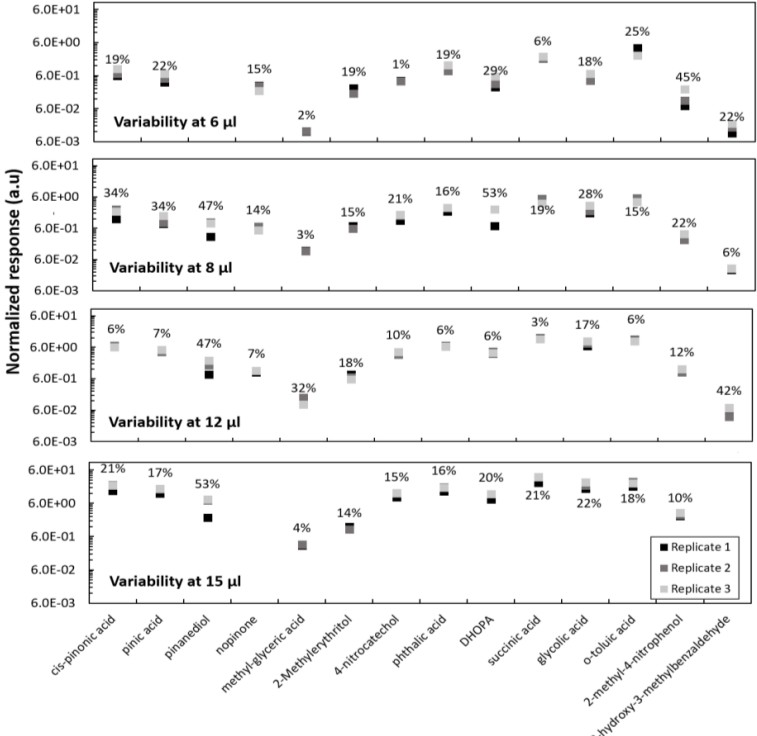

**Figure 4.** Replicates variability at different injection volumes of 6, 8, 12 and 15 µl from a mixture solution of anthropogenic and biogenic standards at 50 mg L$^{-1}$ analysed in the GC-MS. Transition time between replicates injection varies between 3 and 4 days. Compound responses are normalized to the internal standard response.



400 From a mixture of biogenic and anthropogenic standards at 50 µg mL$^{-1}$, volumes of 6, 8, 12 and 15 µL were individually added to quartz filters with a constant volume of 5 µL of a heptanoic acid solution. After filter extraction and analysis, the variability between triplicates for each injection volume was evaluated (Figure 4). Replicates were injected on different days, within 3 and 4 days from the first injection. The injection time was selected as it will be the maximum equivalent time for the injection of real samples, assuring instrumental response and non-significant solution degradation after storage at -18 °C. Longer storage times were not evaluated and are not discussed here.

405 The proposed method seems suitable for most of the target compounds: cis-pinonic acid, pinic acid, pinanediol, α-methylglyceric acid, 2-methylerythritol, 4-nitrocatechol, phthalic acid, DHOPA, succinic acid, glycolic acid, p-toluic acid and 2-methyl-4-nitrophenol. However, for pinanediol and 2-hydroxy-3-hydrobenzaldehyde (Fig. 4), the variability for the injections at 12 and 15 µL was high and the compounds were not detected in some replicates. The lack of detection of those compounds could derive from an incomplete derivatization after 30 mins or enhanced matrix effects 410 at higher concentrations. For other compounds, such as DHOPA and 2-methyl-4-nitrophenol, higher variability was observed at lower measured concentrations close to the LOD.

Quadratic calibrations with $R^2 > 0.90$ were used for the target compounds. The calibration information together with Rt, molecular weight and ions monitored after fragmentation are summarized in Table 3. While quadratic calibrations were more suitable for the GC-MS measurements, the $V_{XO}$ of the method showed higher values than those of the 415 UPLC/ESI-IMS-QTOFMS. The lower performance of this method can be associated to the lower sensitivity. Comparison between both methods will be further discusses in Section 3.3. Coefficients of variation are higher for more substituted compounds such as DHOPA and α-methylglyceric acid and compounds with aldehyde functions (2-hydroxy-3-methylbenzaldehyde), showing the lower performance for those compounds. At contrary, lower values were observed for nopinone (no derivatisation) and acids of lower molecular weight such as glycolic acid and succinic acid. 420 For the rest of the compounds, coefficient of variation was ~40%. The final individual compound concentrations are reported with their experimental error obtained from the quadratic fit for the compound mass and the volume deviation during the sampling.





**Table 3**. Molecular weight (MW) before and after TMS derivatisation, retention time (Rt), limit of detection (LOD), ions monitored and calibration information (a, b, c, $R^2$ and $V_{XO}$). Calibrations represent the normalized response versus the mass deposit on filter. Calibration curves were performed using heptanoic acid as the internal standard at 40 µg mL$^{-1}$ for two mixtures of anthropogenic and biogenic standards at 50 µg mL$^{-1}$ adding 6, 8, 12 and 15 µL on filters. $V_{XO}$ shows the coefficient of variation of the method considering a quadratic calibration.

| Organic compound | MW | TMS MW | Rt (min) | Recovery (%) | LOD (ng) | Ions monitored (m/z) | Calibration curve $ax^2+bx+c$ | | | | |
|---|---|---|---|---|---|---|---|---|---|---|---|
| | | | | | | | a (x10$^{-5}$) | b (x10$^{-3}$) | C (x10$^{-2}$) | $R^2$ | $V_{XO}$ (%) |
| cis-pinonic acid | 184 | 256 | 34.2 | 103 ± 12 | 240 | 73, 83, 171 | 2.9 ± 0.2 | -7 ± 1 | 5 ± 4 | 1 | 152 |
| Pinic acid | 186 | 330 | 37.4 | 112 ± 16 | 380 | 73, 129, 157, 171 | 4 ± 1 | -2 ± 1 | 3 ± 2 | 0.96 | 35 |
| (1S,2S,3R,5S)-(+)-Pinanediol | 170 | 314 | 33.4 | 103 ± 21 | 400 | 73, 130, 183, 198, 299 | 3 ± 2 | -12 ± 11 | 3 ± 3 | 0.91 | 37 |
| 1R-(+)-Nopinone | 138 | - | 21.8 | 104 ± 12 | 37 | 54, 83, 95, 109, 122 | 0.2 ± 0.1 | 0.4 ± 0.8 | 0.7 ± 0.6 | 0.98 | 6 |
| α-methylglyceric acid | 120 | 336 | 28.6 | 94 ± 21 | 140 | 129, 219, 306 | 0.07 ± 0.03 | -0.1 ± 0.2 | 0 | 0.95 | 108 |
| 2-methylerythritol | 136 | 424 | 32.4 | 93 ± 26 | 0.10 | 116, 117, 219 | 0.05 ± 0.1 | 1 ± 0.8 | 0 | 0.95 | 36 |
| 4-nitrocatechol | 155 | 299 | 39.0 | 105 ± 9 | 330 | 73, 284 | 3.1 ± 0.7 | -10 ± 6 | 4 ± 3 | 0.97 | 27 |
| phthalic acid | 166 | 310 | 37.9 | 109 ± 10 | 310 | 73, 147, 295 | 4 ± 1 | -1.3 ± 0.8 | 5 ± 4 | 0.97 | 41 |
| DHOPA | 148 | 364 | 33.8 | 92 ± 5 | 250 | 73, 147, 277, 349 | 2.6 ± 0.6 | -8 ± 5 | 18 ± 10 | 0.98 | 69 |
| succinic acid | 118 | 262 | 28.5 | 109 ± 13 | 320 | 73, 147, 247 | 9 ± 3 | -27 ± 19 | 11 ± 10 | 0.96 | 39 |
| glycolic acid | 76 | 220 | 18.9 | 110 ± 8 | 370 | 73, 147, 205 | 5.5 ± 0.9 | -17 ± 7 | 12 ± 16 | 0.99 | 35 |
| o-toluic acid | 136 | 208 | 28.5 | 105 ± 11 | 200 | 119, 193 | 5 ± 1 | -10 ± 12 | 1 ± 1 | 0.97 | 59 |
| 2-methyl-4-nitrophenol | 153 | 225 | 35.8 | 108 ± 10 | 320 | 165, 210 | 0.7 ± 0.1 | -2 ± 1 | 0.2 ± 0.3 | 0.98 | 30 |
| 2-hydroxy-3-methylbenzaldehyde | 136 | 208 | 31.3 | 98 ± 12 | 280 | 175, 193 | 0.01 ± 0.0 | 1 ± 3 | 0.6 ± 0.4 | 0.99 | 133 |



### 3.3 Methods' validation, application to real samples, and intercomparison

#### 3.3.1 Methods performance comparison

Between the two techniques (UPLC/ESI-IMS-QTOFMS and GC-MS), 23 biogenic and anthropogenic organic markers were quantified, with 5 species being detected by both methods (Table 2 and Table 3). Together, those methods allowed the analysis of a substantial list of aromatic and non-aromatic compounds containing acids, alcohols and aldehyde functions. UPLC/ESI-IMS-QTOFMS offered the advantage of phenol compounds detection at higher sensitivity. For example, UPLC/ESI-IMS-QTOFMS analysis showed lower values on the LODs and recoveries for 4-nitrophenol and 2-methyl-4-nitrophenol, values < 5 ng and > 100%, respectively. This was not the case with the other phenol compound (nitrocatechol), which showed a lower extraction recovery (80%). For the rest of the organic acids and aldehydes, LOD between 23 and 140 ng were observed (Table 2). Lower extraction recoveries were additionally observed for more substituted markers such as MBTCA, terebic acid, syringaldehyde and 2,5-dihydroxy benzoic acid.

As shown in Table 3, LODs for GC-MS were higher than for UPLC/ESI-IMS-QTOFMS, due to its lower sensitivity. Despite the differences in the sensitivity, the extra derivatization step on GC-MS offered the advantage of highly functionalized compounds detection, especially those of higher polarity, enhancing the number of markers that can be quantified. Recovery rates higher than 98% were observed for the biogenic and anthropogenic markers with exception of α-methylglyceric acid, 2-methylerythritol and DHOPA for which recovery rates were lower. For 2-methylerythritol, a polyol with four OH groups susceptible to derivatization with labile protons, the lower recovery rate is attributed to incomplete derivatization, affecting the extraction efficiency. Similarly, α-methylglyceric acid and DHOPA functionalities can derive into three substitutions. Despite the lower LOD values observed by UPLC/ESI-IMS-QTOFMS, a higher recovery rate was observed for 4-nitrocatechol using GC-MS.

LODs of 240 ng for pinonic acid and 380 ng for pinic acid using GC-MS and 63 ng and 44 ng using UPLC/ESI-IMS-QTOFMS were observed. The LODs found in this work are generally higher than the validation parameters for GC-MS previously observed for a comparable method reported by Chiappini et al. (2006). This variability can be attributed to the differences in extraction procedures and derivatisation protocols. Chiappini et al. (2006) performed online SFE, which allows the solvent removal from the separation step, while in this work the presence of the solvent and derivatisation reagent mixture contributes to the background signal, influencing the LOD. It is worth to highlight that Chiappini et al. (2006) validation parameters were provided for the analysis of markers derived from biogenic hydrocarbons while in this work, we provided a wider range of compounds including open ring anthropogenic markers.

Similar LODs for 4-nitrophenol and 2-methyl-4-nitrophenol were observed in this work (17 ng mL$^{-1}$ and 22 ng mL$^{-1}$) compared with those reported by Hoffmann et al. (2007) with values of 27.8 ng mL$^{-1}$ and 13.8 ng mL$^{-1}$, respectively, using HPLC-MS. However, they are higher than those reported by Ikemori et al. (2019) of 1.2 ng mL$^{-1}$ and 0.64 ng



mL$^{-1}$ for 4-nitrophenol and 2-methyl-4-nitrophenol using UPLC-MS/MS. For syringaldehyde, the LOD was one order of magnitude higher than that of Hoffmann et al. (2007) one (707 ng mL$^{-1}$ vs 45.5 ng mL$^{-1}$). For biogenic compounds such as terebic acid, King et al. (2019) provided a method with LOD of 5.7 ng mL$^{-1}$ using a LC-Orbitrap, while here we observed a higher value of 240 ng mL$^{-1}$. Variations between the validation parameters are a consequence of the

sensitivity between the techniques as was also observed by comparing UPLC/ESI-IMS-QTOFMS and GC-MS. The way the LODs were computed for each of the methodologies compared above may also influence the values. LODs of the UPLC/ESI-IMS-QTOFMS presented in this work were higher than LOQ values reported by Albinet et al. (2019) (between 1.0 ng mL$^{-1}$ and 4.0 ng mL$^{-1}$) using HPLC/MSMS. Differences between both studies can derive from sample preparation protocols and the type of calibration, Albinet et al. (2019) reported internal calibrations while in this work

external calibration was performed.

### 3.3.2 Evaluation of UPLC/ESI-IMS-QTOFMS and GC-MS methods on aerosol samples from the Rambouillet forest and intercomparison

Samples collected in the Rambouillet forest were analysed by both UPLC/ESI-IMS-QTOFMS and GC-MS. Some of the biogenic and anthropogenic markers identified are summarized in Figure 5.

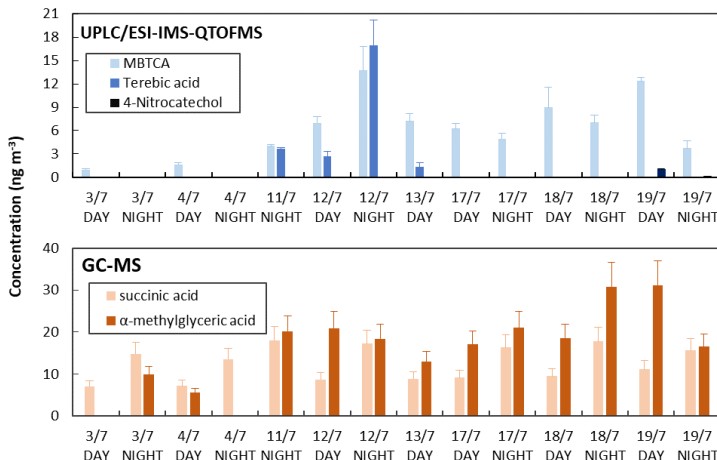


**Figure 5.** Concentrations of a selection of markers of biogenic and anthropogenic origin detected by UPLC/ESI-IMS-QTOFMS (top panel) and GC-MS (bottom panel) analysis of samples collected at the Rambouillet forest (France) during the summer 2022.

Terebic acid was detected in four of the forest samples while MBTCA was detected in most of the samples using

UPLC/ESI-IMS-QTOFMS. For GC-MS analysis anthropogenic and biogenic acids such as succinic and α-





methylglyceric acid were detected. 4-nitrocatechol, which we considered here as a biomass burning marker, was quantified only during July 19, where a fire event was reported (Menut et al., 2023). Five common compounds could be detected: cis-pinonic acid, pinic acid, 4-nitrophenol, 2-methyl-4-nitrophenol, and 4- nitrocatechol. Because the concentrations of the nitro-compounds were below the LOD for GC-MS, hereafter we focus on the comparison of cis-

pinonic acid and pinic acid. As observed in Figure 6, the concentrations of cis-pinonic acid obtained by the two techniques were similar, except for three samples (12/07 NIGHT, 17/07 NIGHT and 18/07 NIGHT) for which the concentrations obtained by UPLC/ESI-IMS-QTOFMS were about double compared to those measured by GC-MS. For the remaining samples, variations were within the measurement uncertainty associated with each technique. For those same samples, concentrations of pinic acid observed by GC-MS were higher than those measured by UPLC/ESI-

IMS-QTOFMS.

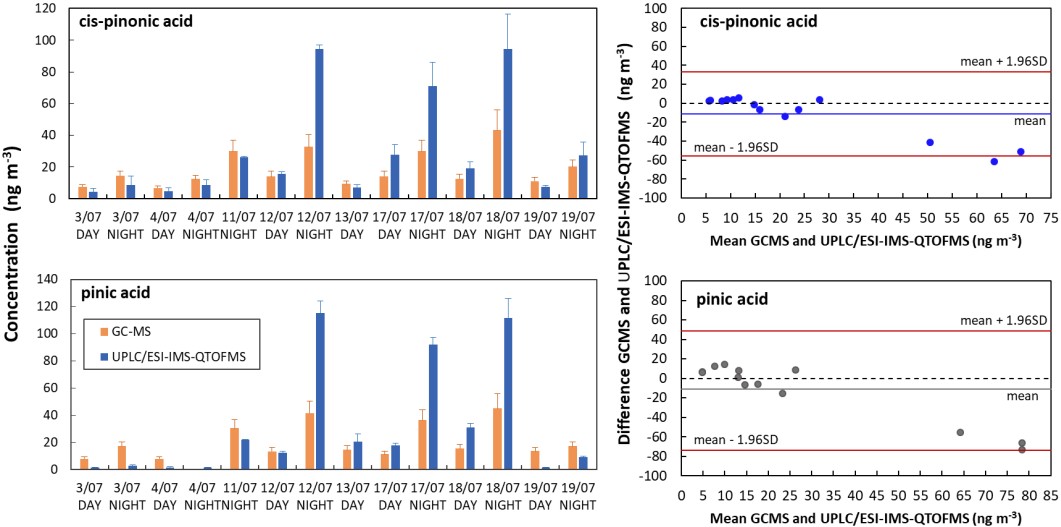

**Figure 6.** Concentrations plots for pinic acid and cis-pinonic acid and Bland-Altman plots for comparison of UPLC/ESI-IMS-QTOFMS and GC-MS methods. Analysis was performed on aerosol samples collected at the Rambouillet forest (France) during the summer 2022. Bland-Altman plots show the difference between UPLC/ESI-

IMS-QTOFMS and GC-MS methods. Black and blue lines show the mean of the difference between measurements and red lines represent the upper and lower limits of agreement, which were calculated considering 1.96 times the standard deviation (Bland and Altman, 1999).

As shown in the Bland-Altman plots (Figure 6), measurements between GC-MS and UPLC/ESI-IMS-QTOFMS are comparable as they fall inside the limits of agreement (red lines). However, at comparing the concentration values, a



mean difference (reported as mean values in Figure 6) between both techniques of 11 ng m$^{-3}$ for cis-pinonic and 13 ng m$^{-3}$ for pinic acid were observed. Both compounds follow a similar behaviour of standard deviation variation on the upper interval, with exception of three measurements, which were systematically closer to the lower limit. Differences between the concentrations observed for both techniques can be influenced by their sensitivity, extraction procedure and sample ageing. As discussed in Section 2.4.1, the derivatization in GC-MS lowers the polarity of the compounds

and influences their detection. Additionally, we assumed that filter samples used here have a homogenous distribution between different pieces used for this analysis, this together with a time of 8 months between UPLC/ESI-IMS-QTOFMS and GC-MS analysis can also introduce discrepancies between the techniques.

## 4 Conclusions

In this paper we describe two complementary methods for the quantification of molecular markers in SOA collected

on filters using UPLC/ESI-IMS-QTOFMS and GC-MS after solvent extraction and derivatisation (for GC-MS). Combining the two methods, the quantification of α/β-pinene, mono-and di-aromatic compounds and a few markers of isoprene oxidation products was possible. We observed high recovery rates (>98%) for most of the organic markers with exception of the most substituted ones for GC-MS (e.g. methylerythritol) and high polar ones such as 2,5-dihydroxy benzoic acid and MBTCA for UPLC/ESI-IMS-QTOFMS. UPLC/ESI-IMS-QTOFMS showed a better

suitability for the analysis of molecular markers, especially for nitro-compounds with LOD< 5 ng, aromatic compounds such as methyl phthalic and salicylic acid (LOD< 30 ng) and less polar biogenic markers such as cis-pinonic and pinic acid. In addition, GC-MS analysis allowed the identification of smaller organic acids and polyols, improving the range of functionalities that can be detected due to the derivatization step. Common compounds comparison derived from both techniques showed a good agreement between different techniques.

**Author contributions**

Conceptualization DLP, AG, ChG, PF; Formal analysis DLP, EM, TB, AG; Funding acquisition AG, PF; Investigation DLP, AG, ChG, PF; Methodology DLP, AG, PF, ChG, EM, TB, CeG; Project administration AG, PF, ChG; Resources AG, PF, ChG; Supervision AG, PF, ChG; Writing original draft DLP, ChG; Writing-review & editing DLP, ChG, AG, PF, EM, TB, CeG, JFD; Final approval of all the authors.

**Declaration of competing interest**

The authors declare no known competing interests that could influence the work reported in this paper.



**Special issue statement.**

This article is part of the special issue "Atmospheric Chemistry of the Suburban Forest – multiplatform observational campaign of the chemistry and physics of mixed urban and biogenic emissions". It is not associated with a conference.

**Acknowledgements**

The authors acknowledge Alexander Albinet for providing the isoprene markers standards and the ACROSS team for their contribution during the samples collection. The authors also acknowledge PRAMMICS Platform from OSU-EFLUVE UMS 3563 for the instrument access.

**Funding sources**

This work was supported by the project TRAC-AOS-A within the LEFE-CHAT national program from CNRS-INSU and from ADEME. The PhD scholarship of DLP is supported by the IDEX program of the Université Paris Cité. The ACROSS project received funding from the French National Research Agency (ANR) under the program ANR−17−MPGA−0002 and supported by the French National program LEFE of CNRS-INSU.

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
