# Peer review of "Two optimized methods for the quantification of anthropogenic and biogenic markers in aerosol samples using liquid chromatography mass spectrometry and gas chromatography mass spectrometry"

_EGUsphere, 2025_

## Author Comment (AC1)

We thank the reviewers for their careful revisions of the manuscript which helped improving the clarity and quality of the text. Please find our point-by-point responses below.

**Reviewer 1**

The authors present two fully characterized complementary methods to quantify SOA marker compounds from filter samples. In this work, a number of target compounds that were previously analyzed in dedicated studies were combined and the instrument response was systematically investigated. The manuscript provides figures of merit for all investigated compounds and the instrument responses were well characterized. Such a combination of methods contributes to the better characterization of SOA composition and the comparable quantification of these marker compounds. The paper is logically structured and well written. I have, however, a few concerns about some of the numbers presented. In my opinion the paper can be suitable for publication in Atmospheric Measurement Techniques once a few issues have been addressed.

1. The authors applied two previously used LC methods and combined both to test a new method. Although the authors claimed that most of the target compounds eluted after 20 min of the second method, the new method was chosen to run for 45 min with a comparable gradient, which seems quite arbitrary. Did the authors test carry-over effects for the shorter LC methods or the effect of longer flushing times on TIC and repeatability?

Most of the standards elutes within 18 min with good peak separation using the chromatographic method of choice with a total duration of 45 min. The method is characterized by a faster elution gradient compared to the 60 min method, with a "cleaning step" at 95% B. We tested carry over effects by injecting blanks in between samples. No carry over was observed.

We have now amended the text at lines 249: "To assess whether carry over would affect the measurements with a reduced method compared with the 60 min one, blanks were injected in between samples. No carry over was observed therefore, the 45 min elution method was selected for the analysis."

2. More to that point, the authors describe in section 3.1.4, that the instrument signal response decreased during a continuous sequence by more that 70%. I am quite concerned about this finding, as the quantification is also depending on this response. Do the authors have an explanation for this behavior and were some parameters tested to improve on that? The authors claim that there is no carry-over effect, as none of the analytes were found in subsequent blank samples, but there seems to be some sort of matrix effect during the ionization. The authors need to provide more details here.

Different experimental parameters were tested to understand whether the origin of this decrease was associated with the stability of the mixture, extraction process or instrumental response. As show in Figure S6, a mixture solution with the target anthropogenic markers was injected into the system by triplicates and there was a decrease between the first and the third injection for all the compounds. We hypothesized that such decrease could be either associated to the stability of the solution or a decreasing instrumental response, as no extraction procedure was applied in this test. We further explored the causes of this loss of signals by comparing injections of the same solutions at different times (Figure S5), as well as injections of calibration (Figure S7) and lock-mass (Figure S8) solutions. We concluded that the signal loss is attributable to a variation in instrumental response rather than instability of the solutions. This is now better explained in the text of the manuscript in sections 3.1.2 to 3.1.4. As the revision was quite extensive, we refer the reviewer to the revised manuscript with tracked changes. This instrument response instability can negatively affect quantification, and this is reflected in the relatively higher LODs obtained for this method compared to some from the literature (Table S4).

We would like to highlight that the drop in sensitivity over time is a known problem with Waters systems(https://support.waters.com/KB\_Inst/Mass\_Spectrometry/WKB62583\_Sensitivity\_drop ping\_over\_time). This is why the software includes an "Automatic Detector Check" option in the monitoring tools, designed to continuously monitor and adjusts the detector voltage to maintain a steady detector gain when running ESI experiments, compensating for sensitivity loss. However, in this work, when fixing the "automatic detector check" tool, no variation was observed.

3. In line 329 the authors describe that for analytes with higher variability, the third replicate was not considered due to loss over time, but it seems to me from figure 2, that due to the loss of signal, the second and third replicate of a sample show lower variability. Can the authors clarify how this effected the analysis?

We agree with the reviewer that due to the signal loss, the variability between the third and second replicates is lower compared to the first replicate. However, we would like to clarify that we used a calibration over time approach for the quantification so that three replicates of the same sample may have a different calibration assigned based on the time of injection

Lines 377-378 were rewritten to clarify this point as follows: For compounds for which the mass derived from the normalized signal present a high variability between the triplicates, only the closest two replicates were considered in the quantification.

4. The GC method applied by the authors was changed from previously used methods. Namely the SPE step was replaced by liquid extraction and direct injection, which resulted in higher LODs compared to the previous method. The authors note that in this work a wider range of compounds were analyzed, but did the authors test the markers investigated in this work with SPE injection?

**Figure R1.** Comparison of identification of DHOPA at using direct injection and SFE extraction on the GC-MS system. Between direct injection and extraction with SFE a 15 mins delay in the Rt derives from the dynamic mode extraction.

We did not use solid phase extraction (SPE) but supercritical fluid extraction (SFE). In this work, the injection of anthropogenic compounds was initially tested with the SFE configuration described by Chiappini et al. (2006). Aiming to follow and quantify markers of monoaromatic compounds (e.g., toluene, xylenes), the detection of 2,3-dihydroxy-4-oxopentanoic acid

(DHOPA) was tested. As observed in Figure R1, the compound was identified through direct injection. However, at performing SFE, the detection of the target marker was not consistent, as its detection was not reproducible between consecutive experiments. The mass spectra resulting from the different derivatized fragments didn't correspond neither to the one of DHOPA. Given the high substitution and polarity of this compound, additional modifiers and higher derivatization times could have been considered, however those changes would have impacted the detection of the other compounds. Therefore, additional extraction protocols, such as the liquid extraction was considered in this work as a compromise for the detection of biogenic and anthropogenic markers.

5. For the GC/MS method, BSTFA is used as a derivatization agent for alcohols. Perhaps the authors could briefly discuss the possibility of using other derivatization agents. This could specifically improve the detection of other compound classes, such as carbonyls.

The following text was added into the manuscript to improve the explanation regarding the derivatizing reagent.

Lines 159-163: BSTFA was selected as one of the most common derivatization reagents for compounds with labile hydrogens (Claeys and Maenhaut, 2021; Cochran et al., 2012; Chiappini et al. 2006) due to the predominance of acidic groups of the target markers (Table 1). However, other derivatization reagents could be used to expand the detection to other functionalities, for example, O-(2,3,4,5,6-pentafluorobenzyl)hydroxylaminehydrochloride for carbonyl compounds (Nozière et al., 2015; Orata, 2012).

6. In the conclusion section the authors compare the LODs for the reported method with literature values. As many previously reported values are much lower, it would be worth highlighting the benefit of this combined method more, to emphasize the value from being able to analyze different compounds classes in one combined protocol.

Following the reviewer's suggestion, we have now highlighted better the advantage of our approach in the conclusion section at line 591 "23 biogenic and anthropogenic molecular markers", at line 594 with five common species (cis-pinonic acid, pinic acid, 4-nitrophenol, 2-methyl-4-nitrophenol, and 4-nitrocatechol)." and lines 595-598 "Additionally, to the best of our knowledge syringaldehyde and terebic acid detection was achieved for the first time using a single analytical method (UPLC/ESI-IMS-QTOFMS). We observed good recovery rates (between 80-130%), determined through filter extraction, for most of the organic markers...".

**Technical comments:**

L.47 should read "nitrooxy organosulfate markers"

The text was modified as indicated by the reviewer.

L.240 The authors should rephrase the last sentence of that paragraph

The text was rephrased as follows:

Lines 265 to 267: Acetonitrile was therefore selected due to the higher elution power observed for the analysis of the target compounds, better suitability for the column (operated at 40 °C) used in this work, and overall better response for most of the target analytes.

L.254 should read "the instrument variability between three randomly injected replicates of the mixture solution without filter extraction was less than 21% for all target compounds."

The text was corrected as suggested by the reviewer.

L.274 should read "Signals were also higher than for sampled using inserts,..."

The text was modified as suggested by the reviewer.

L.277 the authors should emphasize better that this was a hypothetical issue that they tested for and can decline it as a source of variability.

The text was rephrased to clarify this point.

Lines 311 to 315: Variations in sample response when using inserts could be influenced by the amount of sample entering the system. Bubble formation in the bottom of the conical shaped inserts during sample transfer into the insert can limit the sample intake. This hypothesis was discarded as changing the distance of the needle from the bottom of the vial between 5 and 10 mm did not affect the signals. Further explanations for the response variation are explored in section 3.1.4.

L.327 should read "is reported in Table 2"

The text was corrected as suggested by the reviewer.

L.359 unclear reference with "section 0"

The text as corrected as follows: Section 4.2.1

L.381 ff The authors should rephrase that sentence as it is not clear which conditions are referred to

The text was rephrased to clarify this point.

Lines 436 to 438: Although blank contributions to the signal was observed (Fig. 3), most of the target compounds could be identified and quantified. Therefore, a blank filter was simultaneously analyzed with each batch of real samples and its contribution was subtracted.

L.402 I suggest "best representation of real samples" instead of "the maximum equivalent time"

The text was modified as suggested by the reviewer.

L.435 should read "offered the advantage of detecting phenol compounds at higher sensitivity"

The text was modified as indicated by the reviewer.

L.487 I would suggest "were about twice as high as"

The text was modified as indicated by the reviewer.

L.502 The use of "systematically" in the context of three individual samples is a bit unclear

The word systematically was deleted to clarify this point.

**Reviewer 2**

This manuscript presented two analytical methods (UPLC-HRMS and GC-MS) to quantify 23 organic aerosol markers from biogenic and anthropogenic sources, and their application in ambient aerosol samples. The key finding of this study is that the two analytical methods were demonstrated to show complementary capability for comprehensive identification and quantification of many (23 compounds in this study) organic markers with diverse structures that covering both biogenic and anthropogenic sources. Five organic markers were identified by both analytical methods and showed some level of consistency in the ambient aerosol samples.

The text is quite easy to follow. The very unstable response from the UPLC-HRMS instrument is quite surprise for me, which is very usually. If this turns out to be an issue due to poor instrument performance specifically, then the authors should clarify this point and be a bit cautious for generalization. Overall, the two analytical methods used in this study are not new but offer some insights for maximizing the identification capability of atmospheric organic markers and expanding their application, especially for some aerosol samples with mixing emission sources where many targeted organic markers are of interest. I would recommend this manuscript to be published in AMT after the following major concerns are addressed and incorporated, if these comments are helpful for improving the manuscript.

**Major concerns**

1. Usually the recovery range within 80-120% are good and 70-130% are overall acceptable. Can the authors discuss why we need to care about recovery and what are the major factors affecting recovery, and what is the outcome of recovery on quantification and therefore source analysis? By the way, I would suggest to use the ranges (e.g. 80-120% or 70-130%) for recovery throughout the manuscript, rather than recovery above a certain value

We have modified the text as suggested by the reviewer, providing recovery ranges throughout the text of the manuscript. Tables 2 and 3 were modified for consistency and inverted to vertical position as suggested by the editor.

As stated in the main manuscript, recovery information helps discuss the performance of the proposed methods. To improve that description, the following lines were added to discuss the importance of recovery:

Lines 199-202: The recovery not only provides an estimation of trueness of a method (Thompson et al., 2002), but it also defines the extraction efficiency of the target analytes. The recovery can be influenced by parameters such as the analyte concentrations, the matrix, solvent, and extraction procedure (Golubović et al., 2019; Kumar et al., 2022). In this study, the overall recovery of the extraction procedure is calculated.

Lines 500-502: Acceptable values of recoveries are suggested between 70 and 130% (Golubović et al., 2019). The higher ranges of recoveries observed for nitrophenol compounds shows the possible influence of the matrix.

2. Line 65-75 A range of previous literatures were summarized in terms of LODs and recovery, and these are very boring. But the main motivation of this work is still not clear to me, especially I cannot see the uniqueness of this study. I would strongly suggest to add some information in the introduction to address why we need a new method as presented in this study.

We understand that the reviewer may find boring reading LODs and recovery information; however, they provide important information to understand the suitability of a method for the analysis of certain target compounds. Regarding the concern to perform these methods, we have added some information in the introduction and conclusion sections of this work:

Lines 92-93: Additionally, we achieved for the first time quantification of terebic acid and syringaldehyde using a single analytical method (UPLC/ESI-IMS-QTOFMS), unlike previous studies that focused on the detection of only one of the two types of markers.

Lines 594-597: ... with five common species (cis-pinonic acid, pinic acid, 4-nitrophenol, 2-methyl-4-nitrophenol, and 4-nitrocatechol). Additionally, to the best of our knowledge syringaldehyde and terebic acid detection was achieved for the first time using a single analytical method (UPLC/ESI-IMS-QTOFMS). We observed good recovery rates (between 80-130%), determined through filter extraction, for most of the organic markers...

3. The sample extraction and chemical analysis for ambient aerosol samples presented in this study is done without considering the matrix effect. Would matrix effect be considered as a major issue especially the ambient aerosol samples are highly complex mixtures and may affect their LOD, recovery and quantification? Could the calibration curves established from standards are actually not applicable for ambient aerosol samples due to matrix effect?

We understand the reviewer concerns regarding the suitability of the calibration curves. Although the matrix effect can influence the LOD and recovery as observed in the main manuscript, the fact that we use an internal standard for the whole procedure, allows us to correct for it in highly complex samples such as aerosol samples for which the matrix will vary on a sample-to-sample case. Additionally, calibrations presented in this work were computed considering filter extraction as a proxy of effects that the mixtures of compounds collected on those substrates will undergo. For that reason, the mixture of compounds was selected in function of main biogenic and anthropogenic compounds observed in similar areas of analysis. The fact that LC-MS and GC-MS for similar compounds (i.e., cis-pinonic acid and pinic acid) showed comparable values also shows that although of the different matrix influences and protocols of analysis, the quantification is reliable.

4. To minimize the matrix effect, a common pre-treatment for complex samples is to use solid phase extraction (SPE) to remove some inorganics and extremely high molecular weight compounds. Could some efforts be made to compare and evaluate the results for ambient aerosol samples that were treated with and without SPE?

We did not use SPE as it was observed in a previous study from King et al. (2019) that, for some of the target analytes, especially low-molecular weight and high polar, very low recoveries would be obtained. In that study, various cartridges and cleaning/elution protocols were tested and SPE was discarded due to the complete loss of, for example, methyltetrols and very low recoveries for terebic acid.

5. I never work with ion mobility spectra (IMS) but my understanding is that IMS provides another dimension to distinguish isomers. Since UPLC is already able to separate isomers, I wonder whether ion mobility spectra are a bit redundant in this study, especially I only found two isomers among the 23 organic markers. I never found any figure related to IMS, and can any ion mobility spectra be displayed? Even without CCS, relying on RT and m/z is enough for compound identification, isn't it?

We agree with the reviewer that the identification of the target compounds would be possible considering only the Rt and m/z. However, the CCS value provides an extra identification point, thus increasing confidence in the compound detection and identification in complex mixtures such as aerosol samples. In addition, this work presented only a first list of compounds and current work is ongoing to further develop the method including for example, organosulfur compounds and a higher number of biomass burning markers, for which the IMS can provide valuable information for the isomeric separation (Dodds and Baker, 2019). Additionally, CCS

values can contribute to further exploration through non-target analysis on the data already acquired for the ambient samples presented here.

6. Line 210: The purpose of adding 0.1% formic acid can be added and explained. In addition, is mobile A referring to 0.1% formic acid in water and mobile B referring to 0.1% formic acid in methanol? Or mobile A referring to 0.1% formic acid in water and mobile B referring to only methanol? Please clarify.

Formic acid refers to both water and mobile phase B. Line 210 was rewritten to clarify this point and the purpose of adding the formic acid.

Lines 230-233: Mobile phases consisted of 0.1% formic acid (v/v) added in both solvents, ultrapure water (A) and methanol (B). Addition of formic acid is a common practice used to stabilize the solution pH, reducing unwanted adducts formation and leading to the improvement in detection of the analyte signals, ionization, peak shape and separation efficiency (Kaufmann et al., 2024; Liigand et al., 2014).

7. Line 210-220 Two types of extraction solvents (50:50 acetonitrile and water; 50:50 methanol and water) and two types of mobile phase (water + acetonitrile; water + methanol), in total gives a combination of four. Would the extraction solvent (50:50 acetonitrile and water) be more suitable for water plus acetonitrile mobile phase combination rather than water plus methanol?

Thank you for point this out. We would like to clarify that such combination of 4 at mixing solvents was not performed. Mixture of extraction solvent of 50:50 acetonitrile and water were only evaluated with the combination of mobile phases of water/acetonitrile. Similarly for the methanol/water.

We have now clarified this in the text at line 254: "For these tests, standard solutions were prepared in 50/50 ultrapure water/organic solvent mixture, matching the same organic solvent used for mobile phase B in order to minimize any possible artifact that could affect the peak shape even though the injection volume was only 2  $\mu$ L." and at line 265: "Acetonitrile was therefore selected due to the higher elution power observed for the analysis of the target compounds, better suitability for the column (operated at 40 °C) used in this work, and overall better response for most of the target analytes."

8. Line 230 the authors say "As the use of 100% organic solvent to prepare the standard solutions under analysis may negatively influence the peak shape and thus prevents proper quantification, we selected 50/50 ultrapure water/organic solvents for the preparation of standard solutions". My experience is that this depends on injection volume, and injection of pure organic solvent does not affect the peak shape if the injection volume is not very high (e.g.  $5~\mu$ L). In addition, the choice of extraction solvents either pure organic solvent or mixture of water and organic solvent are certainly feasible, which should be primary determined by the soluability and stablility of the sample compounds. Please clarify this point.

We agree with the reviewer that the solvent injected can have little influence provided that the volume injected is small. It is however common practice to inject a solvent composition that is like the composition of the eluent at the time of injection. We have now rephrased the text (line 253) as follow: "For these tests, standard solutions were prepared in 50/50 ultrapure water/organic solvent mixture, matching the same organic solvent used for mobile phase B in order to minimize any possible artifact that could affect the peak shape even though the injection volume was only 2  $\,\mu L$ ."

9. Line 230-235 What is the stability of these organic markers in extracted solvent (50:50)?

The stability of the anthropogenic markers was evaluated by injecting consecutive triplicates of 2 solutions at 1.5 and 2.5  $\mu$ g mL-1 (Table S2). Solutions were prepared on the 50:50 solvent ultrapure water: acetonitrile.

**Table S2.** Stability test performed for the target anthropogenic markers in 50:50 ultrapure water: acetonitrile. Percentage values represent the variability between injections performed by consecutive triplicates.

| Compound name             | 1.5 μg mL -1 | 2.5 μg mL -1 |
|---------------------------|-------------------------|-------------------------|
| 2-methyl-4-nitrophenol    | 6.1%                    | 4.3%                    |
| 3-acetylbenzoic acid      | 2.6%                    | 4.3%                    |
| 4-methylphthalic acid     | 2.8%                    | 9.2%                    |
| 4-nitrocatechol           | 6.0%                    | 5.0%                    |
| 4-nitrophenol             | 6.2%                    | 3.3%                    |
| Syringaldehyde            | 6.8%                    | 11.9%                   |
| 2,5-dihydroxybenzoic acid | 1.9%                    | 15.5%                   |
| Phthalic acid             | 2.9%                    | 6.8%                    |

As observed in the table, the anthropogenic markers are stable in the mixture solvent, as the variability between injections was lower than 10% for most of them at the two concentrations. Similarly for the target biogenic compound as observed in Figure 2. For a higher transition time, the stability of the markers was analyzed by injecting the same three solutions at two different dates (see Figure S5). An increase in the response variability was observed to be compound dependent, with higher values for phenol functionalities at lower concentrations. To avoid possible aging effects or matrix effects of the markers on the extraction solvent, for analysis, samples were extracted, frozen during the night and analyzed in the consecutive date, in a period where the stability of the target compounds was observed (Table S2). Table S2 was added into the supplement of this work, and the following lines on the main text:

Lines 322-328: To better understand the signal decrease of the target compounds, the response from the mixture solutions (Section 3.1.3) of the anthropogenic markers in the 50/50 solvent of ultrapure water/ acetonitrile directly injected (vial) is shown in Fig. S6. Stability test was performed using mixture solutions at 1.5  $\mu$ g mL-1 and 2.5  $\mu$ g mL-1. A decrease between the first and the third injection was observed for all the compounds, which could be either associated to the stability of the solution or a decrease in the instrumental response. The target markers showed to be stable in solution as a variability

Figure R2. Average ratio of the signal compounds with and without extraction.

Regarding the calibrations, they were performed on the spiked filters as we consider they are more representative of the process that the real samples collected will undergo and therefore, it is a better estimation of their concentrations. In the same line, the recoveries were calculated considering the response with the filter extraction for each calibration replicate and averaged for the three replicates. As indicated in the main text; to overcome the problematic signal decrease, each replicate quantification was performed with the closest calibration. An example for the quantification is provided in the reply to question 3 of the reviewer one, to clarify this point.

Regarding the recovery, we refer the reviewer to the main manuscript, where we indicate that in this work the recovery was as a ratio of masses as explained at lines 199-202: "The recovery for each compound for both methods was calculated using the ratio between the amount of the compound found after extraction and the amount added to a filter blank before extraction..."

11. Sect 3.1.3 I feel quite surprised for the large difference with and without insert, as well as the poor linear response as shown in Fig. S4. This usually should not be an issue as it is very common to use an insert in the vial for concentrated sample with low liquid volume. This is probably an issue dependent on specific user and instrument device. Could the material difference (wall of vial vs. insert) affect the ionization? Also, I notice some values are extremely low, is the instrument response in this study referring to fitted peak area from selected ion chromatogram?

We don't believe that the difference between with and without inserts is associated with the material used. We hypothesized that such difference resulted from sample degradation and/or signal loss over time as samples in vials (without inserts) were analyzed first in the sequence. In addition, we considered that because of the use of the conical shape inserts, bubbles could be form in the bottom and influence the sample uptake. To verify this, two distance for the needle position were tested without effect observed. Therefore, we don't consider that the material could affect ionization. The text in section 3.1.3 has now been largely revised. We refer the reviewer to the revised manuscript with tracked changes.

The instrument response corresponds to the sum of the intensities of the peaks of all clusters derived from 3D peak detection

12. Sect 3.1.4 Again, as an experience UPLC-HRMS user, I feel very surprising for the 70% signal loss after 25h for SST solution. This suggests the performance of the instrument is very unstable. This should be very rare and must be dependent by specific instrument. I wonder whether the authors test the standards and between every 24h, since these organic makers especially carboxylic acids are usually stable for at least weeks. Can the authors show time series of individual standards' response over several days by repeating the injection?

The response of individual standards for the calibration tests at different injection dates is summarized in Figure R3 as the average of three injections. Between the different dates, additional concentration points were added to verify linearity in a desired range. Although of the instrumental signal decrease between triplicates injections, the two calibrations are comparable for the target compounds. Strongest differences in the compound responses between the two dates were observed for MBTCA and terebic acid.

The similar compound responses for most of the target markers and the additional observations we have provided in Sections 3.1.3 and 3.1.4, shows that quantification with this instrument is reliable.

Figure R3. Calibrations curves for the target compounds performed at two different dates using the same solutions.

13. Line the authors say "calibrations overtime where performed during analysis to account for the signal stability." UPLC-HRMS can be operated 24/7 and calibration overtime might be a disaster for quantification. Probably this is a poor instrument that should not be used for accurate quantification purpose, I think

We disagree with the reviewer that performing calibration over time "might be a disaster for quantification". Injecting standard solutions throughout a long sequence of analysis ensure that we can monitor any instrument drift and correct for it. Injecting quality control standards would be an alternative approach to achieve the same goal and it is common practice in LC-MS

measurements. In our manuscript, we provided evidence that the quantification is possible, and comparable with the concentrations derived independently from GC-MS analysis.

14. Sec 3.3.1 Those text that summarizing or repeating the LODs are really boring. I would suggest to improve the readability of this section and reduce a bit of the text

Following the reviewer's suggestion, section 3.3.1 was modified as follows:

Lines 513-525: Table S4 summarizes LODs values observed in this work compared to those of the literature for some of the target species. Values for pinonic and pinic acids of this work (> 44 ng) are higher than those reported by Chiappini et al. (2006). This variability can be attributed to differences in the extraction and derivatization steps performed during GC-MS analysis. Chiappini et al. (2006) performed online SFE, which allows the solvent removal from the separation step, while in this work the presence of the solvent and derivatization reagent mixture contributes to the background signal, influencing the LOD. When comparing with Albinet et al. (2019), LODs were compound dependent as similar values were observed for 2-methylerytritol, but not for pinic and cis-pinonic acids, both using GC-MS but different calibration methodologies. Variations were also observed between LC-MS techniques. For example, King et al. (2019) and Amarandei et al. (2023) provided LOD <5.7 ng mL-1 for terebic acid, lower than the one obtained here. Similar LODs for 4-nitrophenol and 2-methyl-4-nitrophenol were observed in this work (17 ng mL-1 and 22 ng mL-1) compared with Hoffmann et al. (2007), but higher than those reported by Ikemori et al. (2019). For syringaldehyde, the LOD was one order of magnitude higher than Hoffmann et al.'s (2007). Such differences among the validation parameters between the different studies can result from instruments sensitivity, sample preparation protocols and calibration types.

Additionally, to improve the readability of the text Table S4 was added into the supplementary.

**Table S4.** LOD comparison of some markers observed in this study with those previously reported in the literature associated with the analysis of aerosol samples.

| Compound name                       | This study LOD          | Literature LOD           | Reference               | Technique   |
|-------------------------------------|-------------------------|--------------------------|-------------------------|-------------|
| cis-pinonic acid b       | 240 ng                  | 6.7 ng                   | Chiappini et al. (2006) | SFE-GC-MS   |
|                                     |                         | 2.2-7.5 ng               | Albinet et al. (2019)   | GC-MS       |
| Pinic acid b             | 380 ng                  | 1.2 ng                   | Chiappini et al. (2006) | SFE-GC-MS   |
|                                     |                         | 6.3-7.6 ng               | Albinet et al. (2019)   | GC-MS       |
| Norpinic acid a          | 190 ng mL -1 | 1.5 ng mL -1  | Amarandei et al. (2023) | LC-MS       |
| Terebic acid a           | 240 ng mL -1 | 5.7 ng mL -1  | King et al. (2019)      | LC-Orbitrap |
|                                     |                         | 0.7 ng mL -1  | Amarandei et al. (2023) | LC-MS       |
| MBTCA a                  | 255 ng mL -1 | 2.7 ng mL -1  | King et al. (2019)      | LC-Orbitrap |
|                                     |                         | 0.9 ng mL -1  | Amarandei et al. (2023) | LC-MS       |
| (1S,2S,3R,5S)-(+)-                  | 400 ng                  |                          |                         |             |
| Pinanediol b             | 400 fig                 |                          |                         |             |
| 1R-(+)-Nopinone b        | 37 ng                   |                          |                         | _           |
| α-methylglyceric acid b  | 560 ng                  | 1.1-2.6 ng               | Albinet et al. (2019)   | GC-MS       |
| 2-methylerytritol b      | 0.1 ng                  | 1.1-4.2 ng               | Albinet et al. (2019)   | GC-MS       |
| 4-nitrocatechol a        | 160 ng mL -1 | 1.0 ng mL -1  | Ikemori et al. (2019)   | LC-MS/MS    |
| Syringaldehydea                     | 707 ng mL -1 | 45.5 ng mL -1 | Hoffmann et al. (2007)  | LC-MS       |
| 4-methyl-phthalic acid a | 150 ng mL -1 | 0.6 ng mL -1  | Ikemori et al. (2019)   | GC-MS       |
| Phthalic acid a          | 44 ng or                | 20 ng                    | Albinet et al. (2019)   | LC-MS/MS    |
|                                     | 220 ng mL -1 | 8.9 ng mL -1  | Amarandei et al. (2023) | LC-MS       |
| DHOPAb                              | 250 ng or               | 1.0 ng mL -1  | Ikemori et al. (2019)   | GC-MS       |

|                                                 | 1000 ng mL -1 | 3.7-11.0 ng                                                               | Albinet et al. (2019)                                                      | GC-MS                      |
|-------------------------------------------------|--------------------------|---------------------------------------------------------------------------|----------------------------------------------------------------------------|----------------------------|
| 2,5-dihydroxy benzoic acid a         | 260 ng mL -1  |                                                                           |                                                                            |                            |
| Succinic acid b                      | 320 ng                   | 1.0-1.3 ng                                                                | Albinet et al. (2019)                                                      | GC-MS                      |
| Glycolic acid b                      | 370 ng                   | 1.6 ng                                                                    | Kitanovski et al. (2011)                                                   | LC-MS                      |
| 3-acetyl-benzoic acid a              | 180 ng mL -1  |                                                                           |                                                                            |                            |
| Salicylic acid a                     | 115 ng mL -1  | 10.2 ng mL -1                                                  | King et al. (2019)                                                         | LC-Orbitrap                |
| o-toluic acid b                      | 200 ng                   |                                                                           |                                                                            |                            |
| 4-Nitrophenol a                      | 17 ng mL -1   | 27.8 ng mL -1 1.2 ng mL -1 0.26 ng mL -1 | Hoffmann et al. (2007)
Ikemori et al. (2019)
Amarandei et al. (2023) | LC-MS
LC-MS/MS
LC-MS |
| 2-methyl-4-
nitrophenol a         | 22 ng mL -1   | 22 ng mL -1
0.64 ng mL -1                        | Hoffmann et al. (2007)
Ikemori et al. (2019)                            | LC-MS
LC-MS/MS          |
| 2-hydroxy-3-
methylbenzaldehyde b | 280 ng                   |                                                                           |                                                                            |                            |

<sup>a when measurements were performed using UPLC/ESI-IMS-QTOFMS, b when measurements were performed using GC-MS

15. Among the 23 organic markers, only 14 were analyzed by UPLC-HRMS. Some other compounds (e.g. o-toluic acid) are expected to be detected by UPLC-HRMS. I wonder whether the authors made some efforts to analyze remaining 9 compounds by UPLC-HRMS?

Thank you for the question. Initial test for the individual injection of the standard compounds for the identification test comprised a higher number of molecular markers, which included some of the other compounds analyzed by GC-MS, such as toluic acid isomer, benzoic acid, DHOPA and nopinone (Figure R4). In the case of nopinone and benzoic acid, those were not detected while toluic acids and DHOPA showed a low compound response. Additionally, for DHOPA, the Rt was <1 min and the CCS value presented a drift time >4 ms, showing it was not well separated and retained by the column. Due to the low compound response of those compounds compare to the rest of the target markers, we decide that analyzing them by GC-MS would be more suitable as the derivatization step will decrease their polarity and favors their detection.

Figure R4. Compound responses for the identification test of biogenic and anthropogenic markers analyzed by means of UPLC/ESI-IMS-QTOFMS. Standards were injected at 10 µg mL-1.

**Minor comments**

1. For table 1, I would suggest to include UPLC and GC info for their target compounds.

The table was modified as indicated by the reviewer.

2. Aromatics are large compound class. Can these aromatic SOA markers in table 1 be assigned to specific aromatic precursor, e.g. toluene and etc?

We have shortened the group to mono-aromatics. We would prefer to let the assignment in general terms as not all the markers belong to a single aromatic precursor. For example, DHOPA is a known marker of the BTEX group (Benzene, Toluene, Ethylbenzene, and Xylenes)(Al-Naiema and Stone, 2017; Srivastava et al., 2023), however more specific isomers such methyl nitrophenols, could not be observed for the same precursors (Forstner et al., 1997).

3. Can Figure 6 also be displayed in 1:1 plot (e.g. x axis vs. y axis referring to UPLC vs. GC method)? By the way, among the five common compounds determined by both methods, only pinic acid and pinonic acid found in ambient aerosol samples?

We have modified Figure 6 following the reviewer suggestion. In addition, we apologize as we found a mistake on the labels of the plot and the representation for pinic acid, those were corrected in the manuscript and the text was modified accordingly. As mentioned in the main manuscript, from the common compounds determined by both methods, only those two were observed. This is not surprising given that the area of analysis is a forested environment (Forest of Rambouillet), from which mainly biogenic markers are expected.

Lines 565 to 570: As observed in Fig. 6, the comparison of the concentration values obtained for cis-pinonic acid and pinic acid showed good determination coefficients (R²> 0.8). For cis-pinonic acid, most of the concentration's values obtained by the two methods are similar, except for three samples for which the concentrations obtained by UPLC/ESI-IMS-QTOFMS were about twice or three times higher than those measured by GC-MS. A similar behavior was observed for pinic acid for the highest concentrations. While for the remaining samples, at lower concentrations, values observed by GC-MS were higher than those measured by UPLC/ESI-IMS-QTOFMS.

4. In Figure S2, it seems MBTCA showing two overlap runs in one figure? I would suggest to label all peaks for Fig. S1-S3 as I found some peaks are not labeled and I guess these could be labeled as impurity?

We have modified the Figures as suggested by the reviewer.

5. Line 140 "section 0" must be a typo

**corrected**

6. In Table S1, I found there are 24 organic makers. Since 23 compounds were studied in the main text, Is there any mistake here? Some text are not displayed in the middle in table S1, which should be fixed

Thank you for pointing this out. As mentioned in the main text (Line 260-264), initial test with the UPLC/ESI-IMS-QTOFMS included the presence of azelaic acid (target compound 24) to discard false positives assignments of the software. However further tests were only performed for the rest of markers.

**References**

Albinet, A., Lanzafame, G. M., Srivastava, D., Bonnaire, N., Nalin, F., and Wise, S. A.: Analysis and determination of secondary organic aerosol (SOA) tracers (markers) in particulate matter standard reference material (SRM 1649b, urban dust), Anal. Bioanal. Chem., 411, 5975–5983, https://doi.org/10.1007/s00216-019-02015-6, 2019.

Al-Naiema, I. M. and Stone, E. A.: Evaluation of anthropogenic secondary organic aerosol tracers from aromatic hydrocarbons, Atmos. Chem. Phys., 17, 2053–2065, https://doi.org/10.5194/acp-17-2053-2017, 2017.

Amarandei, C., Olariu, R. I., and Arsene, C.: Offline analysis of secondary formation markers in ambient organic aerosols by liquid chromatography coupled with time-of-flight mass spectrometry, Journal of Chromatography A, 1702, 464092, https://doi.org/10.1016/j.chroma.2023.464092, 2023.

Chiappini, L., Perraudin, E., Durand-Jolibois, R., and Doussin, J. F.: Development of a supercritical fluid extraction–gas chromatography–mass spectrometry method for the identification of highly polar compounds in secondary organic aerosols formed from biogenic hydrocarbons in smog chamber experiments, Anal. Bioanal. Chem., 386, 1749–1759, https://doi.org/10.1007/s00216-006-0744-3, 2006.

Claeys, M. and Maenhaut, W.: Secondary Organic Aerosol Formation from Isoprene: Selected Research, Historic Account and State of the Art, Atmosphere, 12, 728, https://doi.org/10.3390/atmos12060728, 2021.

Cochran, R. E., Dongari, N., Jeong, H., Beránek, J., Haddadi, S., Shipp, J., and Kubátová, A.: Determination of polycyclic aromatic hydrocarbons and their oxy-, nitro-, and hydroxy-oxidation products, Analytica Chimica Acta, 740, 93–103, https://doi.org/10.1016/j.aca.2012.05.050, 2012.

Dodds, J. N. and Baker, E. S.: Ion Mobility Spectrometry: Fundamental Concepts, Instrumentation, Applications, and the Road Ahead, J. Am. Soc. Mass Spectrom., 30, 2185–2195, https://doi.org/10.1007/s13361-019-02288-2, 2019.

Forstner, H. J. L., Flagan, R. C., and Seinfeld, J. H.: Secondary Organic Aerosol from the Photooxidation of Aromatic Hydrocarbons: Molecular Composition, Environ. Sci. Technol., 31, 1345–1358, https://doi.org/10.1021/es9605376, 1997.

Golubović, J., Heath, E., and Heath, D.: Validation challenges in liquid chromatography-tandem mass spectrometry methods for the analysis of naturally occurring compounds in foodstuffs, Food Chemistry, 294, 46–55, https://doi.org/10.1016/j.foodchem.2019.04.069, 2019.

Hoffmann, D., linuma, Y., and Herrmann, H.: Development of a method for fast analysis of phenolic molecular markers in biomass burning particles using high performance liquid chromatography/atmospheric pressure chemical ionisation mass spectrometry, J. Chromatogr. A, 1143, 168–175, https://doi.org/10.1016/j.chroma.2007.01.035, 2007.

Ikemori, F., Nakayama, T., and Hasegawa, H.: Characterization and possible sources of nitrated mono- and di-aromatic hydrocarbons containing hydroxyl and/or carboxyl functional groups in ambient particles in Nagoya, Japan, Atmos. Environ., 211, 91–102, https://doi.org/10.1016/j.atmosenv.2019.05.009, 2019.

Kaufmann, A., Butcher, P., Maden, K., Walker, S., Widmer, M., and Kaempf, R.: Improved method robustness and ruggedness in liquid chromatography–mass spectrometry by increasing the acid content of the mobile phase, Journal of Chromatography A, 1717, 464694, https://doi.org/10.1016/j.chroma.2024.464694, 2024.

King, A. C. F., Giorio, C., Wolff, E., Thomas, E., Karroca, O., Roverso, M., Schwikowski, M., Tapparo, A., Gambaro, A., and Kalberer, M.: A new method for the determination of primary and secondary terrestrial and marine biomarkers in ice cores using liquid chromatography high-resolution mass spectrometry, Talanta, 194, 233–242, https://doi.org/10.1016/j.talanta.2018.10.042, 2019.

Kitanovski, Z., Grgić, I., and Veber, M.: Characterization of carboxylic acids in atmospheric aerosols using hydrophilic interaction liquid chromatography tandem mass spectrometry, Journal of Chromatography A, 1218, 4417–4425, https://doi.org/10.1016/j.chroma.2011.05.020, 2011.

Liigand, J., Kruve, A., Leito, I., Girod, M., and Antoine, R.: Effect of Mobile Phase on Electrospray Ionization Efficiency, J. Am. Soc. Mass Spectrom., 25, 1853–1861, https://doi.org/10.1007/s13361-014-0969-x, 2014.

Nozière, B., Kalberer, M., Claeys, M., Allan, J., D'Anna, B., Decesari, S., Finessi, E., Glasius, M., Grgić, I., Hamilton, J. F., Hoffmann, T., Iinuma, Y., Jaoui, M., Kahnt, A., Kampf, C. J., Kourtchev, I., Maenhaut, W., Marsden, N., Saarikoski, S., Schnelle-Kreis, J., Surratt, J. D., Szidat, S., Szmigielski, R., and Wisthaler, A.: The Molecular Identification of Organic Compounds in the Atmosphere: State of the Art and Challenges, Chem. Rev., 115, 3919–3983, https://doi.org/10.1021/cr5003485, 2015.

Orata, F.: Derivatization Reactions and Reagents for Gas Chromatography Analysis, in: Advanced Gas Chromatography - Progress in Agricultural, Biomedical and Industrial Applications, edited by: Ali Mohd, M., InTech, https://doi.org/10.5772/33098, 2012.

Srivastava, D., Li, W., Tong, S., Shi, Z., and Harrison, R. M.: Characterization of products formed from the oxidation of toluene and m-xylene with varying NOx and OH exposure, Chemosphere, 334, 139002, https://doi.org/10.1016/j.chemosphere.2023.139002, 2023.

---

## Author Response (AR3)

We thank the reviewers for their careful revisions of the manuscript which helped improving the clarity and quality of the text. Please find our point-by-point responses below.

I want to thank the authors for the taking the time to address all comments. I have, however, still some concerns, that should be addressed prior to publication in AMT. These concern mainly the decreasing signal intensities during a measurement sequence, that were addressed in previous comments Reviewer1-Comment2; Reviewer1-Comment2; Reviewer2-Comments10-13.

1. I still have a hard time judging whether this observation prevents a reliable quantification for samples with variable matrices or not. During consecutive measurements within a sequence, both analyte signals and total signals seem to decrease. The authors describe the effect and did characterization experiments to well characterize the decrease of the signal and correct for it, which is good. But the reproducibility is unclear to me. In Figure S8, the Lockmass signal decreases by a factor of <2 over the course of 19h, but from Figure S7, the same compound signal decreases by a factor of ~5 over 25h. On the other hand, on two different days, the signals seem very comparable, as shown in figure R3 in the authors response as well as in figure S5 of the SI. Are these experiments comparable and what might explain the difference?

To ensure reproducibility of the signal between sequences injected on different days, a source cleaning is performed systematically before each sequence. Source cleaning together with purge and equilibration steps contributes to the signal recovery and comparability as observed in Figure R3. The intensity of the signal is also checked systematically on our reference compound (also called "lockmass"), to ensure there is no issue with the mass spectrometer detection before sample injection. After purging and equilibration, the mass precision of the instrument is assessed via injection of a quality control mixture (SST solution, provided by Waters). Then, the experiments are comparable although of the instrumental signal decrease during the sequence as the signals were further normalized to the internal standard, which accounts for that lost (Figure S9). We believe that the difference highlighted by the reviewer is due to a sensitivity variation of the instrument over time. This is better observed on the following question.

Regarding the concern of the reviewer due to time variations in Figures S7 and S8, we would like to clarify that they belong to different tests with Leucine-Enkephalin. First, the Lockmass, which is a solution of pure Leucine-Enkephalin at 100 pg/µL in a mixture of Acetonitrile/Water/Formic acid 50/50/0.1%, is continuously infused in the source of the mass spectrometer, at 15µL/min, in parallel to the flow coming from the UPLC. Lockmass signal is recorded every 5 minutes for the full length of the analysis sequence and serves as mass calibration correction in real time. The second Leucine-Enkephalin test is part of the SST mixture, which serves as quality control. In this mixture, Leucine-Enkephalin is at 2.5 µg/mL in Water/Acetonitrile/Formic acid 95/5/0.1%. As explained in the main manuscript, Figure S7 represents the signal measure for the SST injected before and after a sequence of 34 samples (approx. 25 hours), while Figure S8 shows the Lockmass signal, which was directly infused in the mass spectrometer.

Lines 146-147 were added in the manuscript: Between sequences, a source cleaning step was performed to increase the instrument sensitivity (as detailed in Section 2.5).

2. It seems to me that the method itself causes the reduction of signal intensity. Have the authors tested the signal intensity with the cleaning step, that was mentioned in the response to Reviewer1-Comment1?

We discarded that the reduction in signal intensity over time was method dependent as such behavior was observed during samples analysis using the method described in the main manuscript, SST injected using a different elution method and the direct infusion of the Lockmass solution. It is an instrumental issue, specific to the Vion series.

[Figure]

Figure R1. Lockmass responses as TIC during sequences injected at two different days (SST sequence then samples) and a source cleaning step in-between.

We have tested the signal variation between different sequences (injection days) for the stability of the Lockmass with a source cleaning step between them. As observed in Figure R1, the signal increases after the cleaning step, without necessarily recovering the same absolute intensity as before and decreases again during the sequence, following a similar behavior at the beginning of each sequence, independently of the total analysis method (extraction, elution, detection). To overcome this, the sequences analysis was selected in a period where we can ensure the instrument response and performed the time dependent calibrations. A new calibration is performed for each sequence.

3. More to that point, how did the authors exclude that the compound-specific decrease of the signal is matrix dependent?

The matrix effect is not excluded in this work, as it can explain for example, the behavior of nitrophenol compounds as highlighted in the manuscript. However, we considered that the signal decrease is not only matrix dependent as during the Lockmass and SST tests the signal decrease was also observed (Question 2). Compound-specific signal variability can also be explained by the differences in ionization efficiencies of different compounds in ESI.

5. Where all measurement sequences performed in the same way, with the samples at the same position in the sequence to apply the correction of the respective replicate or how exactly was the correction factor applied here?

Thank you for point this out. We would like to clarify that there is not a correction factor applied between sequences or samples. During each sequence, the calibration and samples were injected by triplicates. Each measurement was performed in the following order by triplicate: Calibration (from less concentrated to more concentrated) in the same positions and samples (in randomized order). The samples were randomized to take into accounts experimental variability bias.

6. The authors point out, that the drop of sensitivity is a known problem with waters systems, but the provided link refers to an actual technical issue with the instrument that can be addressed and fixed. Was that specific issue occurring with the instrument used in this study and has it been fixed?

Signal decrease over time is indeed a known instrumental issue in Waters MS instruments. A parameter is implemented to correct for this issue for some instruments of Waters (like Synapt, or Xevo, which work with MassLynx software). The link below we provide mentions a software-implemented solution with the addition of the "Automatic Detector Check" function. However we did not observe any improvement of the sensitivity loss when activating the automatic detector check, because the function does not work with our version of QTOF (Vion), as confirmed by Waters Service Engineers. Unfortunately, there is no possible software update that could solve the problem for our version of the QTOF.

https://support.waters.com/KB_Inst/Mass_Spectrometry/WKB99320_How_to_enable_or_disable_Automatic_Detector_Check_on_a_SYNAPT_G2-Si

Minor comments:

I would suggest to label plots in figures with multiple plots to better link the figure caption to the individual plot

Figures and captions were modified in the manuscript as suggested.

L.92 and some more, I would suggest "Time-of-Flight"

Modified as suggested by the reviewer

L.100 should read "first-time"

Corrected

L.333 should read: "Stability tests were performed"

Corrected